# Beyond RLHF and NLHF: Population-Proportional Alignment under an Axiomatic Framework

**Kihyun Kim**[1*]  **Jiawei Zhang**[1,2*†]  **Asuman Ozdaglar**[1]  **Pablo A. Parrilo**[1]

[1] MIT LIDS  [2] University of Wisconsin-Madison

{kihyun, jwzhang, asuman, parrilo}@mit.edu

## Abstract

Conventional preference learning methods often prioritize opinions held more widely when aggregating preferences from multiple evaluators. This may result in policies that are biased in favor of some types of opinions or groups and susceptible to strategic manipulation. To address this issue, we develop a novel preference learning framework capable of aligning aggregate opinions and policies proportionally with the true population distribution of evaluator preferences. Grounded in social choice theory, our approach infers the feasible set of evaluator population distributions directly from pairwise comparison data. Using these estimates, the algorithm constructs a policy that satisfies foundational axioms from social choice theory, namely monotonicity and Pareto efficiency, as well as our newly-introduced axioms of population-proportional alignment and population-bounded manipulability. Moreover, we propose a soft-max relaxation method that smoothly trades off population-proportional alignment with the selection of the Condorcet winner (which beats all other options in pairwise comparisons). Finally, we validate the effectiveness and scalability of our approach through experiments on both tabular recommendation tasks and large language model alignment.

## 1 Introduction

Aligning artificial intelligence (AI) systems with complex human preferences is a growing priority in fields such as robotics (Kupcsik et al., 2017; Bıyık et al., 2024), recommendation systems (Xue et al., 2023), and large language models (LLMs) (Ziegler et al., 2019; Stiennon et al., 2020; Ouyang et al., 2022). A key challenge in this endeavor is how to infer and represent such preferences accurately, particularly when they are only available through incomplete signals like pairwise comparisons. This has prompted reinforcement learning from human feedback (RLHF), which has become a widely used framework for preference learning (Ouyang et al., 2022; Christiano et al., 2017). RLHF streamlines the alignment process by first learning a reward model that assigns scalar scores to different alternatives, typically trained using maximum likelihood estimation under the Bradley–Terry (BT) model. In the second stage, a policy is optimized through reinforcement learning to maximize the expected rewards, guiding the system toward behaviors aligned with human preferences.

Despite its practical success and simplicity, the standard RLHF framework rests on a critical assumption that complex human preferences can be captured by a single scalar reward. Recent research highlights that this assumption often breaks down, especially when human feedback reflects inconsistent or conflicting judgments across evaluators (Chakraborty et al., 2024). In particular, RLHF struggles in scenarios involving intransitive or cyclic preferences, where no clear ranking among alternatives can be established, leading to failures in accurately modeling the underlying preferences (Munos et al., 2024; Swamy et al., 2024). To address these limitations, a game-theoretic framework called Nash learning from human feedback (NLHF) has been introduced (Munos et al., 2024; Swamy et al., 2024; Ye et al., 2024; Maura-Rivero et al., 2025). NLHF reframes preference

---

[*]Equal contribution.

[†]Corresponding author.

learning as a two-player constant-sum game and identifies equilibrium policies that no competing policy can outperform, regardless of the complexity of the underlying preferences.

Nevertheless, both RLHF and NLHF frameworks remain limited in their ability to address another critical issue: the proportional alignment of evaluator preferences. When preferences are aggregated across multiple evaluator groups with distinct viewpoints, both RLHF and NLHF tend to yield policies that do not adequately reflect the full distribution of the evaluator population (Chakraborty et al., 2024). To address these challenges, recent research has turned to social choice theory-oriented approaches, such as maximizing the minimum satisfaction across evaluator groups (Chakraborty et al., 2024; Ramesh et al., 2024) and optimizing social welfare functions (Zhong et al., 2024; Kim et al., 2025). Another line of emerging research, pluralistic alignment (Sorensen et al., 2024), seeks to reflect diverse perspectives in AI systems through approaches such as mixture-based models (Chen et al., 2024), belief-conditioned models (Yao et al., 2025), and steerable models (Adams et al., 2025), with a particular focus on LLMs. However, these methods generally assume explicit knowledge or clear labels of evaluator groups, which limits their practical applicability since group identities are often implicit or unobservable in the real world. Motivated by this limitation, our research aims to achieve proportional alignment without requiring additional information about the evaluator profile.

Our approach builds upon recent works addressing diverse preference aggregation through an axiomatic approach from social choice theory (Mishra, 2023; Siththaranjan et al., 2024; Dai & Fleisig, 2024; Conitzer et al., 2024; Ge et al., 2024; Maura-Rivero et al., 2025; Shi et al., 2025; Xiao et al., 2025). Specifically, we propose a novel preference learning algorithm that satisfies two foundational axioms, monotonicity (ensuring that improving an alternative's ranking cannot decrease its probability) and Pareto efficiency (ensuring that if an alternative is preferred by all, it is favored by the policy), as well as two new axioms we introduce: population-proportional alignment (PPA) and population-bounded manipulability (PBM). The first new axiom, PPA, requires the policy to be at least weakly proportional to evaluator population shares, addressing the insufficient representation of the population distribution under RLHF and NLHF. The second axiom, PBM, bounds the incentive for manipulation as an affine function of the true population share, thereby guaranteeing robustness. Recent studies have highlighted that conventional preference learning methods are susceptible to strategic misreporting (Buening et al., 2025). Unlike existing approaches that incorporate explicit mechanism design to ensure strict strategyproofness (Park et al., 2024; Soumalias et al., 2024; Sun et al., 2024; Hao & Duan, 2025; Buening et al., 2025), our method inherently limits manipulative advantage by constraining policy selection based on estimated feasible population distributions. Further details on related work are provided in Appendix B.

## 1.1 OUR CONTRIBUTION

The first key contribution of this work is demonstrating that the set of feasible population distributions of evaluators can be inferred directly from pairwise comparison data. Leveraging this insight, we develop a novel preference learning framework designed to align policies proportionally with the underlying population distribution. To establish a rigorous theoretical basis, we adopt an axiomatic approach, proving that our framework satisfies two fundamental axioms, monotonicity and Pareto efficiency, and two newly introduced axioms, PPA and PBM. In addition, we propose a novel softmax relaxation method to control the trade-off between proportional alignment and the selection of the Condorcet winner. For practical deployment, we present a scalable algorithm with function approximation, allowing our framework to scale to high-dimensional settings such as LLMs. Finally, the proposed framework is validated through empirical evaluations in both tabular and function approximation settings.

**Organization of the paper.** In Section 2, we formalize the setting of preference learning and probabilistic social choice, and establish connections between them. In Section 3, motivated by a simple negative example, we introduce two desirable axioms alongside two fundamental axioms. In Section 4, we propose a novel preference learning algorithm that satisfies these axioms and provide a theoretical analysis. Finally, Section 5 presents empirical evaluations that demonstrate the effectiveness and scalability of our method. For ease of reference, all mathematical notation used in the paper is summarized in Appendix A.

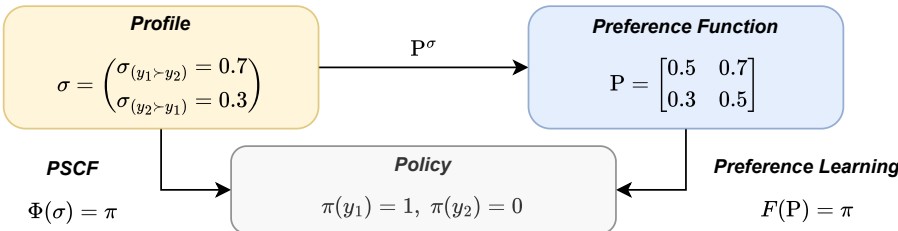

Figure 1: Illustration of the relationships between the profile, preference function, and policy.

## 2 PRELIMINARIES

### 2.1 PROBABILISTIC SOCIAL CHOICE FUNCTION AND PREFERENCE LEARNING

We begin by reviewing key concepts from social choice theory and preference learning to establish a foundation for our subsequent analysis. Consider a set of $M$ alternatives, denoted by $\mathcal{Y} := \{y_1, y_2, \ldots, y_M\}$, where each $y \in \mathcal{Y}$ may represent a response generated by a language model or an action in a decision-making task. We assume each evaluator has a strict and complete ranking over the alternatives, and let $\mathcal{S}$ denote the set of all possible rankings (i.e., permutations of $\mathcal{Y}$). Each ranking is represented by $r \in \mathcal{S}$, where $r(y_i) = k$ indicates that alternative $y_i$ is ranked $k$-th under $r$. A *profile* $\sigma \in \Delta(\mathcal{S})$ is a probability distribution over the set of all rankings, where $\sigma_r$ represents the proportion of the population that adheres to ranking $r$.

A *probabilistic social choice function* (PSCF) is a mapping $\Phi : \Delta(\mathcal{S}) \to \Delta(\mathcal{Y})$ that assigns to each profile $\sigma$ a policy $\pi$, which is a probability distribution over the alternatives in $\mathcal{Y}$. In practice, however, acquiring a complete profile $\sigma$ is often infeasible due to the high cost of collecting full rankings over a large set of alternatives.

To address this limitation, pairwise preference learning algorithms have been developed, allowing alignment based solely on pairwise comparison data. We define a *preference function* $P : \mathcal{Y}^2 \to [0, 1]$, where $P(y \succ y')$ denotes the probability that alternative $y$ is preferred over $y'$. Given a profile $\sigma$, let $P^\sigma$ be the preference function induced by the population distribution $\sigma$ over rankings, defined as

$$P^\sigma(y \succ y') := \sum_{r \in \mathcal{S}} \sigma_r \cdot \mathbf{1}_{\{r(y) < r(y')\}}, \tag{1}$$

where $\mathbf{1}_{r(y) < r(y')}$ is an indicator function equal to $1$ if ranking $r$ places alternative $y$ in a better (i.e., lower) position than $y'$, and $0$ otherwise. This function captures the expected pairwise preference between $y$ and $y'$ under the distribution $\sigma$.

We define $\mathcal{P}$ as the set of all preference functions induced by some profile $\sigma \in \Delta(\mathcal{S})$:

$$\mathcal{P} := \{P \mid \exists \sigma \in \Delta(\mathcal{S}) \text{ s.t. } P = P^\sigma\}. \tag{2}$$

Any $P \in \mathcal{P}$ satisfies consistency conditions known as the Block-Marschak inequalities (Block & Marschak, 1959), including skew-symmetry: $P(y \succ y') + P(y' \succ y) = 1 \ \forall y, y' \in \mathcal{Y}$. A *preference learning algorithm* is a mapping $F : \mathcal{P} \to \Delta(\mathcal{Y})$ that assigns a policy to each preference function. Throughout this paper, we say that a preference learning algorithm $F$ *implements* a PSCF $\Phi$ if, for every profile $\sigma \in \Delta(\mathcal{S})$, it holds that $F(P^\sigma) = \Phi(\sigma)$. The relationships between the profile, preference function, and policy are illustrated in Figure 1.

### 2.2 TWO STANDARD PREFERENCE LEARNING ALGORITHMS

Next, we introduce two prominent preference learning algorithms and discuss their connections to established concepts from probabilistic social choice theory.

**Reinforcement learning from human feedback (RLHF).** The Bradley–Terry (BT) model, widely used in preference modeling, assigns each alternative $y_i$ a reward $r_i$ with preference probabilities

$P(y_i \succ y_j) = \exp(r_i)/(\exp(r_i) + \exp(r_j))$. Standard RLHF estimates these rewards by likelihood maximization and then trains a policy to maximize expected rewards. Recent work (Siththaranjan et al., 2024) shows that this procedure is equivalent to the *maximal Borda rule* from social choice theory, which deterministically chooses the alternative with the highest Borda score $B(y) := \sum_{r \in \mathcal{S}} \sigma_r(M - r(y))$. As proved in Appendix C, the ranking from BT-optimized rewards coincides with Borda rankings, so RLHF without regularization (denoted by $F^{\mathrm{RL}}$) implements the maximal Borda rule (denoted by $\Phi^{\mathrm{MB}}$). Direct preference optimization (DPO) generalizes this by adding Kullback–Leibler (KL) regularization relative to a reference policy (Rafailov et al., 2023).

**Nash learning from human feedback (NLHF).** As highlighted in recent studies (Munos et al., 2024; Swamy et al., 2024; Maura-Rivero et al., 2025), RLHF has limitations in scenarios involving intransitive or cyclic preferences. An alternative $y^*$ is called a *Condorcet winner* if it is preferred by a majority over every other alternative, formally stated as $P(y^* \succ y) > 0.5$ for all $y \neq y^*$. When aggregating preferences across multiple evaluators, scenarios without a Condorcet winner can arise, which is called the *Condorcet paradox*. In such cases, selecting the alternative with the highest Borda score fails to adequately represent collective preferences, as a deterministic policy cannot capture the lack of consensus or nuanced preferences. To address intransitive preferences, the game-theoretic approach, known as Nash learning from human feedback (NLHF) (Munos et al., 2024; Swamy et al., 2024; Ye et al., 2024; Maura-Rivero et al., 2025), has been adopted to model preference learning as a two-player constant-sum game $\max_{\pi_1 \in \Delta(\mathcal{Y})} \min_{\pi_2 \in \Delta(\mathcal{Y})} \mathbb{E}_{(y_1, y_2) \sim (\pi_1, \pi_2)}[P(y_1 \succ y_2)]$, where the equilibrium policy $\pi^*$ cannot be uniformly outperformed. This algorithm, denoted by $F^{\mathrm{NL}}$, implements the well-known PSCF *maximal lotteries (ML)* (Fishburn, 1984), denoted by $\Phi^{\mathrm{ML}}$.

# 3 AXIOMATIC FRAMEWORK FOR POPULATION-PROPORTIONAL ALIGNMENT

## 3.1 MOTIVATING EXAMPLE WITH BINARY ALTERNATIVES

Despite their practical utility, neither RLHF nor NLHF guarantees alignment proportional to the evaluator's preferences. To illustrate this point, we present a simple scenario involving binary alternatives. Consider two alternatives, $\mathcal{Y} = \{y_1, y_2\}$, and a profile $\sigma$ consisting of two distinct groups of evaluators: group $G_1$ prefers alternative $y_1$ over $y_2$, while group $G_2$ prefers $y_2$ over $y_1$. Let $w_1^\sigma$ and $w_2^\sigma$ denote the population shares of groups $G_1$ and $G_2$, respectively. Suppose the two alternatives are nearly tied, with $(w_1^\sigma, w_2^\sigma) = (1/2 + \epsilon, 1/2 - \epsilon)$ for an arbitrarily small positive scalar $\epsilon$. Then, the corresponding preference function is given by $P^\sigma(y_1 \succ y_2) = 1/2 + \epsilon$ and $P^\sigma(y_2 \succ y_1) = 1/2 - \epsilon$. Despite this minimal margin $\epsilon$, both algorithms $F^{\mathrm{RL}}(P^\sigma)$ and $F^{\mathrm{NL}}(P^\sigma)$ yield a deterministic policy that selects the alternative with slightly greater support, namely $y_1$, because such subtle differences in preferences (or rewards) are lost during the policy optimization.

This binary example highlights two potential limitations of RLHF and NLHF frameworks. First, selecting policies that focus entirely on a single alternative may not accurately represent preferences across evaluators, raising concerns about bias. Second, these methods have high sensitivity to small perturbations in the preference function. Specifically, a slight shift in $\epsilon$ from negative to positive abruptly flips the policy outcome $(\pi(y_1), \pi(y_2))$ from $(0, 1)$ to $(1, 0)$, making such approaches vulnerable to small perturbations. These limitations underscore the need for a novel approach that reflects the ratio of $(w_1^\sigma, w_2^\sigma)$ in the resulting policy.

## 3.2 PROPOSED AXIOMS FOR POPULATION-PROPORTIONAL ALIGNMENT AND ROBUSTNESS

Social choice theory studies the aggregation of individual preferences through an *axiomatic* approach, which specifies desirable properties (axioms) and characterizes aggregation rules that satisfy them. In particular, two fundamental axioms, *monotonicity* and *Pareto efficiency*, are presented in Appendix D. Following this approach, we introduce the axioms that a PSCF $\Phi$ is desired to satisfy and propose a preference learning algorithm $F$ that implements such a PSCF.

**Proposed axioms.** Motivated by the earlier example, we next introduce a new axiom designed to ensure alignment with population distribution of preferences. Let $G_k := \{r \in \mathcal{S} \mid r(y_k) = 1\}$ denote the set of rankings in which alternative $y_k$ is ranked first. The population share of group $G_k$ is denoted by $w_k^\sigma := \sum_{r \in G_k} \sigma_r$. For notational convenience, we define $\sigma_k \in \Delta(\mathcal{S})$ as the normalized

Table 1: Overview of standard PSCFs and axioms

| $\Phi$ | $F$ | Monotonicity | Pareto Efficiency | PPA | PBM |
|---|---|---|---|---|---|
| Maximal Borda (MB) | ✓ (RLHF) | ✓ | ✓ | × | × |
| Maximal lotteries (ML) | ✓ (NLHF) | × | ✓ | × | × |
| Random dictatorship (RD) | × | ✓ | ✓ | ✓ | ✓ |
| Proposed framework | ✓ | ✓ | ✓ | ✓ | ✓ |

sub-profile restricted to rankings in $G_k$, where $\sigma_{k,r} = \sigma_r/w_k^\sigma$ for all $r \in G_k$, and $\sigma_{k,r} = 0$ for all $r \notin G_k$. Let $\mathrm{P}_k^\sigma$ denote the group-specific preference function, generated from $\sigma_k$, using the mapping defined in equation 1. By construction, $\mathrm{P}_k^\sigma(y_k \succ y) = 1$ for all $y \neq y_k$, since this group unanimously prefers $y_k$ over all other alternatives. The overall preference function is then a weighted aggregation of the group-specific preferences: $\mathrm{P}^\sigma = \sum_{k=1}^{M} w_k^\sigma \mathrm{P}_k^\sigma$.

Under this definition, our first axiom ensures that the policy reflects each group's population share. Note that our proportionality notion focuses solely on the selection probability of each group's top choice and does not incorporate lower-ranked preferences.

**Definition 3.1** ($\alpha$-Population-proportional alignment ($\alpha$-PPA)). A PSCF $\Phi$ satisfies $\alpha$-*population-proportional alignment* if $\pi(y_k)/w_k^\sigma \geq \alpha(\sigma)$ for all $\sigma \in \Delta(\mathcal{S})$ and $y_k \in \mathcal{Y}$, where $\pi = \Phi(\sigma)$ and $\alpha : \Delta(\mathcal{S}) \to (0, 1]$.

The function $\alpha(\sigma)$ quantifies the strength of alignment: a higher value of $\alpha$ implies stronger alignment with $w^\sigma$, with $\alpha(\sigma) = 1$ indicating perfect proportional alignment. Next, we examine the robustness of $\Phi$ against manipulation through the following axiom.

**Definition 3.2** (Single-group manipulated profile). Given a profile $\sigma$ and a group index $k \in [M]$, a profile $\sigma_k'$ is called a *single-group manipulated profile* of $\sigma$ if $\sigma_k'$ can be obtained by modifying only the ranking distribution of the sub-profile $\sigma_k$. Formally, $\sigma_k'$ is a single-group manipulated profile of $\sigma$ if there exists a profile $\sigma'$ such that $\sigma_k' = \sigma + w_k^\sigma(\sigma' - \sigma_k)$.

**Definition 3.3** ($\gamma$-Population-bounded manipulability ($\gamma$-PBM)). A PSCF $\Phi$ satisfies $\gamma$-*population-bounded manipulability* if, for any profile $\sigma$ and its single-group manipulated profile $\sigma_k'$, we have $\Phi(\sigma_k')(y_k) \leq \gamma_1 w_k^\sigma + \gamma_2$, where $\gamma = (\gamma_1, \gamma_2)$, $\gamma_1 > 0$, and $\gamma_1 + \gamma_2 = 1$.

The $\gamma$-PBM axiom ensures that the maximum influence a single group can exert through manipulation is bounded above by an affine function of its population share. Specifically, a group can only achieve a deterministic policy selection for its preferred alternative (i.e., $\Phi(\sigma_k')(y_k) = 1$) only if it constitutes the entire evaluator population (i.e., $w_k^\sigma = 1$). Note that a larger $\gamma_1$ provides a stronger robustness guarantee. Particularly, $\gamma_1 = 1$ implies that the manipulated policy value is limited exactly to the group's true population share. We also note that the focus of the $\gamma$-PBM axiom differs from that of classical strategyproofness: it does not constrain an individual participant's incentive to misreport, but instead limits the extent to which any group can become over-represented.

### 3.3 LIMITATIONS OF STANDARD PSCFs: AXIOM VIOLATIONS AND NON-IMPLEMENTABILITY

We next show that the standard PSCFs either fail to satisfy the proposed axioms or are not implementable by a preference learning algorithm. Consider a PSCF that aligns the policy exactly with each group's population distribution, commonly referred to as a *random dictatorship* (Brandt, 2017).

**Definition 3.4** (Random dictatorship). A PSCF $\Phi^{\mathrm{RD}}$ is called a *random dictatorship* if $\Phi^{\mathrm{RD}}(\sigma) = w^\sigma$ for all $\sigma \in \Delta(\mathcal{S})$.

By definition, $\Phi^{\mathrm{RD}}$ satisfies both proposed axioms in their strongest forms: $\alpha$-PPA with $\alpha(\sigma) = 1$ for all $\sigma \in \Delta(\mathcal{S})$, and $\gamma$-PBM with $\gamma = (1, 0)$. The following proposition establishes that $\Phi^{\mathrm{MB}}$ and $\Phi^{\mathrm{ML}}$ violate even the weakest forms of these axioms, whereas $\Phi^{\mathrm{RD}}$ satisfies all four axioms.

**Proposition 3.5.** $\Phi^{\mathrm{MB}}$ *and* $\Phi^{\mathrm{ML}}$ *violate the $\alpha$-PPA axiom for any $\alpha$ and the $\gamma$-PBM axiom for any $\gamma$.* $\Phi^{\mathrm{RD}}$ *satisfies all four axioms.*

The proof is provided in Appendix E. Unfortunately, $\Phi^{\mathrm{RD}}$ is not implementable by any pairwise preference learning algorithm, since distinct profiles $\sigma_1$ and $\sigma_2$ may induce identical preference functions $\mathrm{P}^{\sigma_1} = \mathrm{P}^{\sigma_2}$ but different population distributions $w^{\sigma_1} \neq w^{\sigma_2}$ (see Appendix F for an example). Because $w^\sigma$ cannot be recovered solely from $\mathrm{P}^\sigma$, no mapping from preference functions to policies can implement $\Phi^{\mathrm{RD}}$[1]. Our goal, therefore, is to construct a preference learning algorithm $F$ that implements a PSCF $\Phi$ satisfying all four axioms. Table 1 summarizes the standard PSCFs, their implementability, and satisfaction of the four axioms; see Brandl et al. (2022) for additional details.

## 4 ALGORITHMIC FRAMEWORK AND THEORETICAL GUARANTEES

### 4.1 POPULATION DISTRIBUTION RECOVERY FROM PAIRWISE PREFERENCES

In this section, we introduce a preference algorithm $F$, which implements a PSCF satisfying all four axioms presented in the previous section. The framework first estimates the feasible set of underlying population distributions $w$ from given pairwise preferences $\mathrm{P}$, and subsequently constructs a policy $\pi$ closely aligned with the inferred feasible set. We begin with the definition of a feasible population distribution and the characterization of the set of all feasible population distributions.

**Definition 4.1.** A population distribution $w$ is considered *feasible* given $\mathrm{P}$, if there exists a profile $\sigma \in \Delta(\mathcal{S})$ such that $w = w^\sigma$ and $\mathrm{P} = \mathrm{P}^\sigma$.

**Proposition 4.2.** *The set of all feasible population distributions given $\mathrm{P}$ can be expressed as*

$$\mathcal{W}(\mathrm{P}) := \left\{ w \in \Delta(\mathcal{Y}) \,\middle|\, \exists (\mathrm{P}_1, \dots, \mathrm{P}_M) \in \mathcal{P}^M \ s.t. \ \mathrm{P} = \sum_{i=1}^M w_i \mathrm{P}_i, \right.$$
$$\left. \mathrm{P}_i(y_i \succ y) = 1 \ \forall y \in \mathcal{Y} \setminus \{y_i\}, \ \forall i \in [M] \right\}. \tag{3}$$

See Appendix G for the proof. In words, a population distribution $w$ is feasible if and only if there exist group-specific preference functions $(\mathrm{P}_1, \dots, \mathrm{P}_M)$ such that $\mathrm{P}$ is their weighted aggregation, and each $\mathrm{P}_i$ reflects a group of evaluators who unanimously prefer $y_i$ over all other alternatives.

The exact characterization of the set $\mathcal{W}(\mathrm{P})$ is challenging due to the constraints imposed by the set $\mathcal{P}$. We therefore propose a tractable polyhedral outer approximation of the set $\mathcal{W}(\mathrm{P})$, with the number of constraints growing only linearly with the dimension $M$.

**Definition 4.3.** For each $i \in [M]$, define $u_i := \min_{y \in \mathcal{Y} \setminus \{y_i\}} \mathrm{P}(y_i \succ y)$.

**Theorem 4.4.** *The set of feasible population distributions satisfies*

$$\mathcal{W}(\mathrm{P}) \subseteq \overline{\mathcal{W}}(\mathrm{P}) := \left\{ w \in \Delta(\mathcal{Y}) \,\middle|\, w_i \leq u_i \ \forall i \in [M] \right\}. \tag{4}$$

The proof is given in Appendix H. To provide intuition, note that $u_i = 1 - \max_{y \neq y_i} \mathrm{P}(y \succ y_i) = 1 - \mathrm{P}(y' \succ y_i)$, where $y'$ is the alternative most preferred over $y_i$. Thus, $u_i$ represents the remaining population share after excluding those who prefer $y'$ to $y_i$. Thus, $w_i$ cannot exceed this value, as the $w_i$ proportion of evaluators would always report $y_i$ as their preferred option. The tightness of the outer approximation is further discussed in Appendix I, and the relation between Theorem 4.4 and Tatli et al. (2024) is examined in Appendix J. Since $w^\sigma$ is not identifiable from pairwise comparison data, perfect proportional alignment (i.e., $\alpha(\sigma) = 1$ for all $\sigma \in \Delta(\mathcal{S})$) is fundamentally unattainable. Moreover, even achieving a uniform guarantee $\alpha(\sigma) > 2/M$ for all $\sigma$ is impossible for any preference learning algorithm (see Appendix K). This motivates designing algorithms that achieve $\alpha$-PPA with the largest possible $\alpha$.

### 4.2 PROPOSED ALGORITHMIC FRAMEWORK WITH AXIOMATIC GUARANTEES

Given a polyhedron $\overline{\mathcal{W}}(\mathrm{P})$, our goal is to select a policy $\pi$ that guarantees the proportional alignment to all $w \in \overline{\mathcal{W}}(\mathrm{P})$. To this end, we propose to assign probabilities to alternatives in proportion to the derived upper bounds $u_i$.

---

[1]In the literature, the class of implementable PSCFs is often referred to as the C2 class (Fishburn, 1977)

**Definition 4.5.** The preference learning algorithm $F^*$ maps a preference function P to the policy

$$\pi(y_i) = \frac{u_i}{\sum_{j=1}^{M} u_j} \quad \forall i \in [M]. \tag{5}$$

Let $\Phi^*$ denote the PSCF implemented by $F^*$.

This construction adopts a conservative strategy for handling uncertainty in $w^\sigma$ by assigning probabilities proportional to the most conservative estimate of each $w_i^\sigma$. By doing so, the algorithm minimizes the worst-case misalignment caused by the inevitable information loss from pairwise comparisons. Formally, it solves $\max_{\pi \in \Delta(\mathcal{Y})} \min_{w \in \overline{\mathcal{W}}(\mathrm{P})} \|\pi/w\|_\infty$.

We first establish the foundational axiomatic guarantees of the proposed framework.

**Theorem 4.6** (Monotonicity & Pareto efficiency)**.** *The proposed PSCF $\Phi^*$ satisfies the monotonicity and the Pareto efficiency.*

The proofs are provided in Appendix L. Next, we show that $\Phi^*$ satisfies the $\alpha$-PPA axiom. The following lemma establishes that the ratio between the resulting policy and the true population share is lower bounded by the inverse of the total sum of the upper bounds $u_i$.

**Lemma 4.7.** *For any profile $\sigma \in \Delta(\mathcal{S})$, the policy $\pi = \Phi^*(\sigma)$ satisfies*

$$\frac{\pi(y_i)}{w_i^\sigma} \geq \left( \sum_{j=1}^{M} u_j \right)^{-1} \quad \forall i \in [M]. \tag{6}$$

The next lemma shows that this lower bound depends on the number of non-dominated alternatives:

**Definition 4.8** ($\delta$-dominated alternative)**.** *For any $\delta \in [0,1]$, an alternative $y \in \mathcal{Y}$ is said to be $\delta$-dominated in a profile $\sigma$ if there exists an alternative $y' \in \mathcal{Y} \setminus \{y\}$ such that $\mathrm{P}^\sigma(y' \succ y) \geq \delta$.*

**Lemma 4.9.** *Let $w^{\sigma,1}$ and $w^{\sigma,2}$ denote the largest and second-largest elements of $w^\sigma$, respectively. Consider any $\delta \in [0,1]$, and let $N_\delta^\sigma$ be the number of alternatives that are not $\delta$-dominated in profile $\sigma$. Then the lower bound in equation 6 lies within the range $[\alpha(\sigma), 1]$, where*

$$\alpha(\sigma) := \left[ (N_\delta^\sigma - 1)(1 - w^{\sigma,1}) + (1 - w^{\sigma,2}) + (M - N_\delta^\sigma)(1 - \delta) \right]^{-1}. \tag{7}$$

See Appendix M for the proofs. Combining both Lemmas, we obtain the following $\alpha$-PPA guarantee:

**Theorem 4.10** ($\alpha$-PPA)**.** *The PSCF $\Phi^*$ satisfies the $\alpha$-PPA axiom with $\alpha$ defined in equation 7.*

Lemma 4.7 suggests that the actual alignment performance improves as $\sum_{j=1}^{M} u_j$ approaches 1. This typically occurs when the number of non-1-dominated alternatives is small. Notably, when there are only two non-1-dominated alternatives, substituting $N_1^\sigma = 2$ and $w^{\sigma,1} + w^{\sigma,2} = 1$ into equation 7 yields $\alpha(\sigma) = 1$, implying the perfect PPA in such cases. Moreover, when there exists a single dominating group, meaning $(w^{\sigma,1}, w^{\sigma,2})$ approaches $(1, 0)$, then $\alpha(\sigma)$ also approaches 1. Importantly, because $\sum_{j=1}^{M} u_j$ can be computed directly from a given preference function P, the alignment accuracy of the resulting policy can be evaluated at test time. In Appendix N, we present additional bound analysis under a random-ranking model with a practically relevant level of variance. Specifically, we report the average lower bounds from Lemma 4.7 and Lemma 4.9, showing that both yield meaningful guarantees.

Finally, we present the population-bounded manipulability of the proposed method.

**Theorem 4.11** ($\gamma$-PBM)**.** *Let $\pi' = \Phi^*(\sigma_k')$ denote a policy resulting from single-group manipulation by group $G_k$. Then, the following inequality holds:*

$$\pi'(y_k) \leq \frac{u_k}{u_k + 1 - w_k^\sigma} \leq \frac{1}{2}(w_k^\sigma + 1). \tag{8}$$

*Thus, the PSCF $\Phi^*$ satisfies $\gamma$-PBM with $(\gamma_1, \gamma_2) = (1/2, 1/2)$.*

The proof is provided in Appendix O. Note that $(\gamma_1, \gamma_2) = (1/2, 1/2)$ represents a worst-case bound. The actual manipulability for each group is more tightly bounded by $u_k/(u_k + 1 - w_k^\sigma)$. For instance, if $u_k \leq 1/2$ and $w_k^\sigma \leq 1/2$, then $\pi'(y_k) \leq 1/2$. This indicates that a non-majority group cannot elevate their preferred alternative to majority status through manipulation. In addition, the above result can be interpreted as a weaker form of strategyproofness (see Appendix P for details).

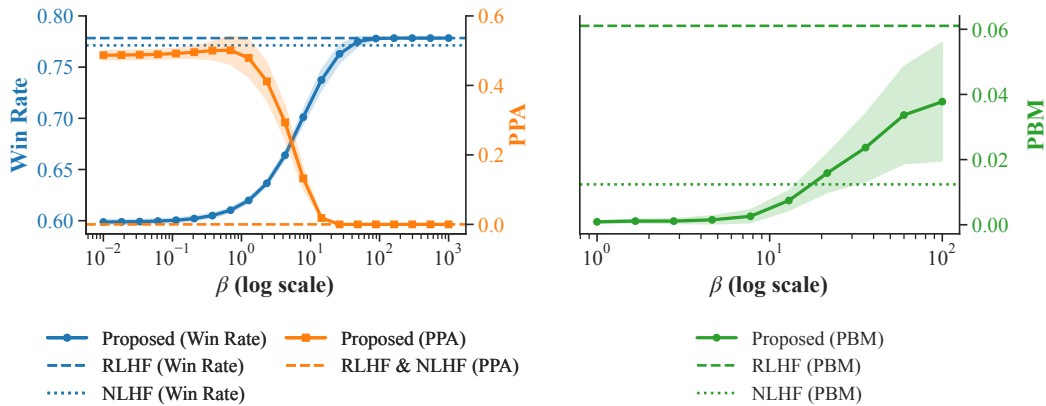

Figure 2: Tabular experiment results (Section 5.1) for $F^\beta$, $F^{\text{RL}}$, and $F^{\text{NL}}$. **Left:** win rate (left axis, blue) and PPA level (right axis, orange). **Right:** PBM level (policy gain through manipulation).

### 4.3 BALANCING PPA AND CONDORCET CONSISTENCY

While $F^*$ and $\Phi^*$ are deliberately designed to satisfy PPA, one may still wish to incorporate majority-based principles such as Condorcet consistency. However, it is impossible for any method to simultaneously satisfy both $\alpha$-PPA and Condorcet consistency.

**Definition 4.12** (Condorcet consistency). A PSCF $\Phi$ satisfies *Condorcet consistency* if, for any profile $\sigma$ with a Condorcet winner $y^*$, $\Phi(\sigma)(y^*) = 1$.

**Proposition 4.13.** *No PSCF can simultaneously satisfy $\alpha$-PPA and Condorcet consistency.*

See Appendix Q for the proof. To balance two axioms, we propose a softmax-relaxed algorithm $F^\beta$ (and its corresponding PSCF $\Phi^\beta$), by modifying $F^*$ as follows:

$$\pi(y_i) = \frac{u_i \exp(\beta u_i)}{\sum_{j=1}^{M} u_j \exp(\beta u_j)} \quad \forall i \in [M]. \tag{9}$$

The parameter $\beta \geq 0$ controls how sharply the policy concentrates on alternatives with higher $u_i$ values. When $\beta = 0$, the algorithm reduces to the original $F^*$. As $\beta \to \infty$, the policy becomes deterministic and converges to $\pi(y^*) = 1$, where $y^* = \arg\max_{i \in [M]} u_i$. This limiting $\Phi^\infty$ is the well-known minimax Condorcet method (Kramer, 1975), which satisfies Condorcet consistency (see Appendix R for the proof).

**Proposition 4.14.** $\Phi^\infty$ *satisfies Condorcet consistency.*

The softmax relaxation offers a smooth trade-off between $\alpha$-PPA and Condorcet consistency, controlled by the parameter $\beta$. We analyze the theoretical behavior of intermediate $\beta$ values in Appendix S, and empirically demonstrate the effects of varying $\beta$ in Section 5. Additionally, Appendix T discusses the connection to pairwise majority consistency (PMC) (Ge et al., 2024), which imposes a stronger consistency requirement, ensuring the entire policy ranking aligns with majority preferences.

## 5 EXPERIMENTS

### 5.1 TABULAR EXPERIMENT: MOVIE RECOMMENDATION

**Datasets and experimental setup.** To validate our theoretical findings, we evaluate the framework on a movie recommendation task using 1,297 evaluator rankings over 20 movies from the MovieLens 1M dataset (Harper & Konstan, 2015). In each episode, we sample $10^5$ pairwise comparisons i.i.d. from the true preference function $\mathrm{P}^\sigma$ and train $F^\beta$ alongside two baselines, $F^{\text{RL}}$ and $F^{\text{NL}}$.

We report averages and standard deviations over 50 episodes on three metrics: (i) win rate against a uniform policy, $\mathbb{E}_{(y_1, y_2) \sim (\pi, U)}[\mathrm{P}^\sigma(y_1 \succ y_2)]$, where $U$ is the uniform distribution over $\mathcal{Y}$, (ii) PPA level, $\alpha(\sigma) = \min_{i \in [M]} \pi(y_i)/w_i^\sigma$, and (iii) PBM, the average policy gain from a single group's strategic manipulation.

Table 2: Win rate and PPA level $\alpha(\sigma)$ across datasets and algorithms

| Dataset | Category | Metric | $\beta = 0$ | $\beta = 10^{-4}$ | $\beta = 10^{-2}$ | $\beta = 10^0$ | DPO |
|---|---|---|---|---|---|---|---|
| Synthetic | Color | Win rate | 0.6157 | 0.6880 | 0.6961 | 0.8429 | **0.8566** |
| | | PPA ($\alpha$) | **0.0883** | 0.0235 | 0.0183 | 0.0003 | 0.0000 |
| Alpaca-GPT4 | Expertise | Win rate | 0.7613 | 0.7610 | 0.7634 | 0.7636 | **0.7697** |
| | | PPA ($\alpha$) | **0.1428** | 0.1418 | 0.1392 | 0.1273 | 0.1321 |
| | Style | Win rate | 0.8398 | 0.8432 | 0.8425 | **0.8530** | 0.8478 |
| | | PPA ($\alpha$) | **0.5012** | 0.4197 | 0.3637 | 0.3635 | 0.3786 |

**Results and discussion.**    As shown in the left panel of Figure 2, RLHF and NLHF achieve high win rates of 0.7784 and 0.7712, respectively, but both yield a PPA level of 0. For our proposed algorithm $F^\beta$, we observe the expected trade-off: as $\beta$ increases, the win rate rises from 0.5987 to 0.7784, while the PPA level decreases from 0.4869 to 0. These results confirm our theoretical prediction of each algorithm's behavior. Additionally, the average value of $u_i$ across $i$ was 0.1892, suggesting that the set $\underline{\mathcal{W}}(\mathrm{P})$ in equation 4 provides a meaningfully tight estimate of $w^\sigma$ in our method.

Regarding PBM, the average gain was calculated as 0.0611 for RLHF, 0.0124 for NLHF, and $8.896 \times 10^{-4}$ when $\beta = 10^0$. Overall, $F^\beta$ outperforms the baselines when $\beta \le 10^1$, indicating that our proposed algorithm significantly reduces susceptibility to manipulation and supports its robustness guarantee.

## 5.2   Large-scale experiment: instruction-tuned LLMs

**Datasets and experimental setup.**    We next evaluate the algorithm in high-dimensional settings with function approximation by fine-tuning the Qwen2.5-3B-Instruct model (Yang et al., 2024). For a synthetic dataset, we construct 10 questions asking evaluators which color they prefer, with 10 candidate colors as possible responses. The true rankings of 1,000 evaluators are generated from randomly sampled rewards, and $10^4$ pairwise comparisons are drawn i.i.d. from $\mathrm{P}^\sigma$. We next test the algorithm on the Alpaca-GPT4 dataset (Peng et al., 2023), which contains 52k prompts. Following prior work (Jang et al., 2023; Chakraborty et al., 2024), we consider two group categories (expertise and style) and sample one pairwise comparison per prompt using GPT-4.1 (Achiam et al., 2023). Further details on data generation and hyperparameters are provided in Appendix U.

For both datasets, we evaluate two metrics: (i) the win rate against a reference policy (the pretrained model), $\mathbb{E}_{(x,y_1,y_2)\sim(\rho,\pi,\pi_{\mathrm{ref}})}[\mathrm{P}^\sigma(y_1 \succ y_2 \mid x)]$, and (ii) the PPA level $\alpha(\sigma)$, comparing the results with DPO as the baseline. To estimate the output policy (i.e., the group distribution of generated responses), we used response logits directly for the synthetic dataset, and group classifications from the annotation model (GPT-4.1) for the Alpaca-GPT4 dataset. The specific training algorithm is described in Appendix V, and the full experimental code is included in the supplemental material.

**Results and Discussion.**    Table 2 presents the win rate and PPA level $\alpha(\sigma)$ across datasets and algorithms. On the synthetic dataset, we observe a clear trade-off between win rate and PPA, confirming that $\beta$ effectively controls this balance and validating the algorithm's effectiveness in high-dimensional settings. For the Alpaca–GPT4 dataset, the trade-off is present but less pronounced, largely because group distributions are inferred via an annotation model (GPT-4.1), which introduces noise and obscures the effect of $\beta$. In contrast, the synthetic dataset allows direct computation from response logits, enabling more precise estimates. These results suggest that a small synthetic dataset can be used to evaluate a model's PPA level and tune $\beta$ to reach a desired target.

We highlight several practical considerations for deployment. First, our two-phase function approximation approach (learning $u$ and $\pi$), has computational cost comparable to RLHF and higher than DPO, suggesting the need for direct policy-optimization methods. Second, accurately evaluating PPA levels in LLMs remains an open challenge, extending beyond the two approaches we consider, namely logit-based estimation and group classification. As this paper primarily introduces the theoretical framework with supporting experiments, our findings should be viewed as initial evidence of scalability, with further algorithmic and evaluation advances expected to strengthen these results.

## 6 CONCLUSION AND FUTURE DIRECTIONS

This paper introduces a novel preference-learning framework that aligns policies proportionally with population distributions inferred from pairwise comparison data. We believe this framework offers a new perspective on alignment algorithms by shifting the focus beyond the conventional emphasis on win rate. Furthermore, our work strengthens the connection between preference learning and social choice theory by implementing a new class of probabilistic social choice functions, extending beyond standard rules such as maximal Borda and maximal lotteries. Future research will aim to extend the framework to incorporate lower-ranked preferences and to develop more efficient algorithms for high-dimensional environments.

## ACKNOWLEDGMENTS

This work was supported by the Office of Naval Research grant N000142512296. Jiawei Zhang was supported by the Office of the Vice Chancellor for Research and Graduate Education at the University of Wisconsin–Madison with funding from the Wisconsin Alumni Research Foundation. The authors acknowledge the MIT Office of Research Computing and Data for providing high performance computing resources that contributed to the results reported in this paper.

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

# A    NOTATION

We summarize the mathematical notation used in the paper.

| Symbol | Description |
|---|---|
| *Rankings, profiles, groups, and preferences* | |
| $[M]$ | The set of integers $\{1, 2, \ldots, M\}$. |
| $\mathcal{Y} = \{y_1, \ldots, y_M\}$ | The set of $M$ alternatives. |
| $\Delta(\mathcal{Y})$ | Probability simplex over a finite set $\mathcal{Y}$. |
| $\mathcal{S}$ | The set of all rankings (permutations) over $\mathcal{Y}$. |
| $r \in \mathcal{S}$ | A ranking, where $r(y_i) = k$ means $y_i$ is ranked $k$-th. |
| $\sigma \in \Delta(\mathcal{S})$ | A profile, i.e., population distribution over rankings. |
| $\sigma_r \in [0, 1]$ | Proportion of evaluators who adopt ranking $r$. |
| $G_k$ | Group $k$, set of rankings where $y_k$ is ranked first. |
| $w_k^\sigma \in [0, 1]$ | Population share of evaluators whose top choice is $y_k$. |
| $\sigma_k \in \Delta(\mathcal{S})$ | Sub-profile of group $G_k$ (evaluators who rank $y_k$ first). |
| $\pi \in \Delta(\mathcal{Y})$ | A policy, i.e., probability distribution over alternatives. |
| $\mathrm{P} \in \mathcal{P}$ | Preference function, $\mathrm{P}(y \succ y')$ is the probability $y$ is preferred to $y'$. |
| $\mathrm{P}^\sigma \in \mathcal{P}$ | Preference function induced by a profile $\sigma$. |
| $\mathrm{P}_k, \mathrm{P}_k^\sigma$ | Group-specific preference function for $G_k$. |
| $\mathcal{P}$ | Set of all preference functions induced by some profile in $\Delta(\mathcal{S})$. |
| *Preference learning algorithms, PSCFs, and axioms* | |
| $F : \mathcal{P} \to \Delta(\mathcal{Y})$ | Preference learning algorithm, mapping a preference function to a policy. |
| $\Phi : \Delta(\mathcal{S}) \to \Delta(\mathcal{Y})$ | Probabilistic social choice function (PSCF), mapping a profile to a policy. |
| $F^{\mathrm{RL}}, \Phi^{\mathrm{MB}}$ | RLHF algorithm and its PSCF (maximal Borda rule). |
| $F^{\mathrm{NL}}, \Phi^{\mathrm{ML}}$ | NLHF algorithm and its PSCF (maximal lotteries). |
| $B(y)$ | Borda score: $B(y) := \sum_{r \in \mathcal{S}} \sigma_r \left( M - r(y) \right)$. |
| $\alpha(\sigma) \in \mathbb{R}$ | Strength of population-proportional alignment (PPA) guarantee. |
| $\gamma = (\gamma_1, \gamma_2) \in \mathbb{R}^2$ | Parameters characterizing population-bounded manipulability (PBM). |
| $\sigma_k'$ | Single-group manipulated profile of $\sigma$ (group $k$ perturbs only its sub-profile). |
| $u_i \in [0, 1]$ | $u_i := \min_{y \neq y_i} \mathrm{P}(y_i \succ y)$, upper bound on feasible population share for $y_i$. |
| $\mathcal{W}(\mathrm{P})$ | Set of feasible population distributions consistent with preference function $\mathrm{P}$. |
| $\overline{\mathcal{W}}(\mathrm{P})$ | Polyhedral outer approximation of $\mathcal{W}(\mathrm{P})$ (via $w_i \leq u_i$ constraints). |
| $\delta \in [0, 1]$ | Dominance threshold (used in $\delta$-domination definition). |
| $N_\delta^\sigma$ | Number of alternatives not $\delta$-dominated under profile $\sigma$. |
| $w_1^\sigma, w_2^\sigma$ | Largest and second-largest elements of $w^\sigma$. |
| $y^*$ | Condorcet winner satisfying $\mathrm{P}(y^* \succ y) > \frac{1}{2}$ for all $y \neq y^*$. |
| $F^*, \Phi^*$ | Proposed (baseline) algorithm/PSCF using $u_i$ with $\pi(y_i) \propto u_i$. |
| $F^\beta, \Phi^\beta$ | Softmax-relaxed algorithm/PSCF with concentration parameter $\beta \geq 0$. |
| *Offline learning algorithm with function approximation* | |
| $x \in \mathcal{X}$ | Context (e.g., prompt or state) and context space. |
| $\mathcal{D} = \{(x_i, y_i^w, y_i^\ell)\}_{i=1}^N$ | Offline dataset of pairwise comparisons ($y^w$ preferred to $y^\ell$). |
| $\rho(x), \pi_d(y \mid x)$ | Context (prompt) and query data distribution. |
| $\mu$ | Selector model used to form $u$. |
| $\mathcal{F}_\mu, \mathcal{F}_\pi$ | Function classes for $\mu$ and $\pi$. |
| $\widehat{\mathrm{P}}, \widehat{\mu}, \widehat{u}$ | Empirical estimate of $\mathrm{P}$, $\mu$, and $u$. |
| $\widehat{\pi}_\beta, \widehat{\pi}$ | Softmax policy constructed from $\widehat{u}$ and final estimated policy |

## B    ADDITIONAL RELATED WORK

In this section, we discuss recent work that aims to proportionally represent the diversity of human preferences.

**Limitations of the BT model.**    Recent studies have highlighted limitations of the standard RLHF approach under BT model assumption, which fails to capture the multifaceted and sometimes conflicting nature of human preferences. For example, Kim et al. (2024) demonstrated that the standard MLE algorithms under the BT model can become unstable, particularly in the presence of evaluators exhibiting greedy behavior. They proposed to address this limitation by estimating a set of feasible reward functions without relying on specific modeling assumptions. Additionally, Siththaranjan et al. (2024) established a theoretical equivalence between RLHF and the Borda voting rule, showing that the optimized rankings from standard methods frequently violate majority preferences. To address this issue, they introduced a distributional approach incorporating hidden context variables to address diverse evaluator preferences. Furthermore, Ge et al. (2024) analyzed reward optimization methods under parameterizations, revealing their inherent violation of fundamental axioms such as Pareto efficiency. They proposed a novel algorithm explicitly designed to satisfy these axioms.

**Approaches from social choice theory.**    Parallel research efforts have explored unbiased aggregation of heterogeneous human preferences, grounded in social choice theory. Chakraborty et al. (2024) formally proved the impossibility of equitably aligning single-reward models across diverse evaluator groups, and proposed learning reward mixtures using the EM algorithm followed by maximizing the minimum utility across subpopulations. Additionally, Zhong et al. (2024) conducted a rigorous analysis of multi-group reward learning under various social welfare criteria, such as Nash, utilitarian, and Leximin functions, and provided theoretical alignment guarantees. Park et al. (2024) proposed a probabilistic opinion pooling function that directly aggregates multiple probabilistic models into a single policy, as well as personalized algorithms that output individualized policies after estimating confidence sets. Shi et al. (2025) analyze the theoretical limits of NLHF, showing that exact preference matching is generally impossible, highlighting intrinsic limitations of this paradigm. Concurrent work by Xiao et al. (2025) is closely related to our work. They investigate the tension between RLHF's empirical success and its incompatibility with social choice axioms (PMC and Condorcet consistency), showing that RLHF can satisfy them under a practical assumption about preference labeling. Moreover, they propose a new axiom, *group preference matching*, which requires the policy to reproduce group-level preference distributions in proportion to their population weights. However, they do not provide an algorithmic framework that satisfies this axiom.

**Proportional representation in voting systems.**    The concept of proportional representation has been extensively studied through an axiomatic lens within voting systems. A foundational axiom in multi-winner voting systems is *proportionality for solid coalitions* (PSC), which dictates that any solid coalition (a group of voters who agree on their preferred set of winners) must be guaranteed a number of elected candidates proportional to its population size (Dummett, 1984). Building on this, work in approval-based voting introduced *justified representation* (JR) and its stronger variant, *extended justified representation* (EJR), which ensure that every cohesive group (voters who approve the same set of candidates) receives proportional representation (Aziz et al., 2017). Proportional representation has also been widely applied to *participatory budgeting* (PB) (Aziz & Shah, 2020), focusing on axiomatic methods for distributing funds among public projects under a budget constraint. More recently, Ebadian et al. (2024) analyze distortion bounds of voting rules (Procaccia & Rosenschein, 2006) under a proportional fairness objective. However, the literature defining these proportional representation notions typically assumes either approval-based multi-winner elections (Aziz et al., 2018) or access to full preference information such as ordinal rankings Aziz & Lee (2021); Peters et al. (2021); Airiau et al. (2023); Ebadian et al. (2024). This stands in sharp contrast to our approach, which operates under the minimal assumption of pairwise comparison data and seeks a probabilistic choice distribution over candidates, rather than a fixed multi-winner.

**Pluralistic alignment.**    Emerging research on pluralistic alignment seeks to reflect diverse perspectives in AI systems, with a particular focus on LLMs. Sorensen et al. (2024) outlined three complementary frameworks for pluralistic alignment: Overton pluralism, which captures the range of reasonable responses; steerable pluralism, which allows models to adapt to particular attributes; and

distributional pluralism, which aligns model outputs with population-level distributions. Chen et al. (2024) introduced a framework that modeled heterogeneous human preferences from the ground up using the ideal point model and mixture modeling. Yao et al. (2025) proposed group distributional preference optimization (GDPO), a method that aligns models with the group preferences by estimating the underlying belief distribution and conditioning responses on those beliefs, ensuring representation of both majority and minority views. Adams et al. (2025) developed a steerable pluralistic alignment algorithm, enabling models to adapt to individual preference profiles through few-shot comparative regression across fine-grained attributes. While these approaches show promise, they generally rely on explicit group identification, restricting their applicability in scenarios where group labels are unavailable or difficult to determine. In contrast, our work does not require explicit knowledge of evaluator groups. Instead, we infer population distributions directly from pairwise comparison data and align policies accordingly. In parallel with our work, the problem of recovering a distribution over reward functions from aggregate pairwise comparisons is formalized under the pairwise calibration framework (Halpern et al., 2025). Our approach can be interpreted as selecting a structured solution within this inherently ill-conditioned problem, further constrained to satisfy the proportional alignment and robustness properties that we target.

## C  EQUIVALENCE OF BT-MLE REWARDS RANKING AND BORDA RANKING

**Proposition C.1.** *Let $r^* \in \mathbb{R}^M$ be a maximizer of the likelihood function*

$$L(r) := \sum_{i<j} \left[ \mathrm{P}^\sigma(y_i \succ y_j) \log\left(\frac{e^{r_i}}{e^{r_i} + e^{r_j}}\right) + \mathrm{P}^\sigma(y_j \succ y_i) \log\left(\frac{e^{r_j}}{e^{r_i} + e^{r_j}}\right) \right]. \quad (10)$$

*Then, the ordering of alternatives induced by $r^*$ is identical to the ordering induced by the Borda score $B$ of $\sigma$. Formally, for any $i, j \in [M]$,*

$$r_i^* > r_j^* \quad \Longleftrightarrow \quad B(y_i) > B(y_j). \quad (11)$$

*Proof.* The gradient of $L(r)$ with respect to $r_i$ is given by:

$$\frac{\partial L(r)}{\partial r_i} = \sum_{j \neq i} \left[ \mathrm{P}^\sigma(y_i \succ y_j) - \mathrm{sigmoid}(r_i - r_j) \right], \quad (12)$$

where $\mathrm{sigmoid}(x) := 1/(1 + e^{-x})$. At the optimal solution $r^*$, the first-order condition requires that

$$\sum_{j \neq i} \left[ \mathrm{P}^\sigma(y_i \succ y_j) - \mathrm{sigmoid}(r_i^* - r_j^*) \right] = 0. \quad (13)$$

Now, consider two distinct alternatives $i$ and $k$, and suppose that $r_i^* > r_k^*$. Since the sigmoid function is monotonically increasing, for any $j \neq i, k$, we have $\mathrm{sigmoid}(r_i^* - r_j^*) > \mathrm{sigmoid}(r_k^* - r_j^*)$, and also $\mathrm{sigmoid}(r_i^* - r_k^*) > \mathrm{sigmoid}(r_k^* - r_i^*)$. From the first-order conditions at optimality, we have:

$$\sum_{j \neq i} \mathrm{P}^\sigma(y_i \succ y_j) = \sum_{j \neq i} \mathrm{sigmoid}(r_i^* - r_j^*) \quad \text{and} \quad \sum_{j \neq k} \mathrm{P}^\sigma(y_k \succ y_j) = \sum_{j \neq k} \mathrm{sigmoid}(r_k^* - r_j^*). \quad (14)$$

Since $r_i^* > r_k^*$, it follows that

$$\sum_{j \neq i} \mathrm{sigmoid}(r_i^* - r_j^*) > \sum_{j \neq k} \mathrm{sigmoid}(r_k^* - r_j^*). \quad (15)$$

Therefore, we have

$$\sum_{j \neq i} \mathrm{P}^\sigma(y_i \succ y_j) > \sum_{j \neq k} \mathrm{P}^\sigma(y_k \succ y_j). \quad (16)$$

By definition, $\mathrm{P}^\sigma(y_i \succ y_j) = \sum_{r \in \mathcal{S}} \sigma_r \cdot \mathbf{1}_{\{r(y_i) < r(y_j)\}}$. Substituting this into the inequality above, we get

$$\sum_{r \in \mathcal{S}} \sigma_r \cdot \sum_{j \neq i} \mathbf{1}_{\{r(y_i) < r(y_j)\}} > \sum_{r \in \mathcal{S}} \sigma_r \cdot \sum_{j \neq k} \mathbf{1}_{\{r(y_k) < r(y_j)\}}. \quad (17)$$

Recall that the Borda score is defined as $B(y) := \sum_{r \in \mathcal{S}} \sigma_r \cdot (M - r(y))$, we can rewrite the inner sums in the inequality as:

$$\sum_{j \neq i} \mathbf{1}_{\{r(y_i) < r(y_j)\}} = (M - 1) - (r(y_i) - 1) = M - r(y_i), \tag{18}$$

and similarly,

$$\sum_{j \neq k} \mathbf{1}_{\{r(y_k) < r(y_j)\}} = M - r(y_k). \tag{19}$$

Thus, the inequality becomes

$$\sum_{r \in \mathcal{S}} \sigma_r \cdot (M - r(y_i)) > \sum_{r \in \mathcal{S}} \sigma_r \cdot (M - r(y_k)), \tag{20}$$

which is equivalent to $B(y_i) > B(y_k)$.

For the converse, assume $B(y_i) > B(y_k)$. Following similar steps in reverse, this implies

$$\sum_{j \neq i} \mathrm{P}^\sigma(y_i \succ y_j) > \sum_{j \neq k} \mathrm{P}^\sigma(y_k \succ y_j), \tag{21}$$

which leads to

$$\sum_{j \neq i} \mathrm{sigmoid}(r_i^* - r_j^*) > \sum_{j \neq k} \mathrm{sigmoid}(r_k^* - r_j^*). \tag{22}$$

This inequality can only hold if $r_i^* > r_k^*$. Therefore, we have shown that $r_i^* > r_j^* \iff B(y_i) > B(y_j)$, completing the proof. $\square$

## D   FUNDAMENTAL AXIOMS: MONOTONICITY AND PARETO EFFICIENCY

In this section, we present the definition of two fundamental axioms in social choice theory: *monotonicity* and *Pareto efficiency*. For detailed discussions of these axioms, we refer readers to Brandt (2017); Ge et al. (2024).

**Definition D.1** (Monotonicity). A PSCF $\Phi$ satisfies monotonicity if, for any alternative $y \in \mathcal{Y}$, improving its ranking in a profile without changing other relative rankings cannot decrease its probability in the resulting policy. Formally, if profile $\sigma'$ is obtained from $\sigma$ by improving the ranking of $y$ in some $r \in \mathcal{S}$ with $\sigma_r > 0$, then $\Phi(\sigma')(y) \geq \Phi(\sigma)(y)$.

**Definition D.2** (Pareto efficiency). A PSCF $\Phi$ satisfies *Pareto efficiency* if, whenever an alternative $y$ is ranked above $y'$ in every ranking $r$ with nonzero population share, the resulting policy assigns at least as much probability to $y$ as to $y'$. Formally, if $r(y) < r(y')$ for all $r \in \mathcal{S}$ with $\sigma_r > 0$, then $\Phi(\sigma)(y) \geq \Phi(\sigma)(y')$.

## E   PROOF OF PROPOSITION 3.5

We first demonstrate that $\Phi^{\mathrm{MB}}$ and $\Phi^{\mathrm{ML}}$ violate $\alpha$-PPA. Consider a preference profile $\sigma$ with the following characteristics: (i) The population share of each group is nearly identical, with $G_1$ having a population share $w_1^\sigma$ that is $\epsilon$ greater than the average, and $G_2$ having a population share $w_2^\sigma$ that is $\epsilon$ less than the average. (ii) Within each group $G_k$, there is indifference between any two alternatives other than $y_k$. That is, $\mathrm{P}_k^\sigma(y_i \succ y_j) = 1/2$ for all $i, j \neq k$. Given this profile, we will show that for any $\epsilon > 0$, both RLHF and NLHF yield a deterministic policy that selects $y_1$.

The population distribution and pairwise preference function satisfy

$$w^\sigma = \left( \frac{1}{M} + \epsilon, \frac{1}{M} - \epsilon, \frac{1}{M}, \frac{1}{M}, \dots, \frac{1}{M} \right) \text{ and } \mathrm{P}_k^\sigma(y_i \succ y_j) = \frac{1}{2}, \quad \forall i, j \neq k. \tag{23}$$

Then, the aggregated pairwise preferences $\mathrm{P}^\sigma$ are computed as follows:

- $\mathrm{P}^\sigma(y_1 \succ y_2) = \frac{1}{2} + \epsilon$
- $\mathrm{P}^\sigma(y_1 \succ y) = \frac{1}{2} + \frac{\epsilon}{2}$ for any $y \neq y_1, y_2$

- $\mathrm{P}^\sigma(y_2 \succ y) = \frac{1}{2} - \frac{\epsilon}{2}$ for any $y \neq y_1, y_2$
- $\mathrm{P}^\sigma(y \succ y') = \frac{1}{2}$ for any $y, y' \neq y_1, y_2$

Under this profile, for any $\epsilon > 0$, both $\Phi^{\mathrm{MB}}$ and $\Phi^{\mathrm{ML}}$ result in a policy where $\pi(y_1) = 1$, and $\pi(y_i) = 0$ for any $i \neq 1$. This implies that $\pi(y_i)/w_i^\sigma = 0$ for any $i \neq 1$, which violates $\alpha$-PPA for any $\alpha > 0$.

Next, we show that $\Phi^{\mathrm{ML}}$ violates $\gamma$-PBM using the profile described earlier with $M = 3$. The aggregated preference function $\mathrm{P}^\sigma$ can be represented by the following matrix:

$$\mathrm{P}^\sigma = \begin{bmatrix} \frac{1}{2} & \frac{1}{2} + \epsilon & \frac{1+\epsilon}{2} \\ \frac{1}{2} - \epsilon & \frac{1}{2} & \frac{1-\epsilon}{2} \\ \frac{1-\epsilon}{2} & \frac{1+\epsilon}{2} & \frac{1}{2} \end{bmatrix}. \tag{24}$$

Now, suppose that group $G_3$ manipulates their sub-profile from $\mathrm{P}_3^\sigma(y_1 \succ y_2) = \frac{1}{2}$ to $\mathrm{P}_3^{\sigma'}(y_1 \succ y_2) = 0$. Then, the resulting manipulated aggregated preference function $\mathrm{P}^{\sigma'}$ is calculated as:

$$\mathrm{P}^{\sigma'} = \begin{bmatrix} \frac{1}{2} & \frac{1}{3} + \epsilon & \frac{1+\epsilon}{2} \\ \frac{2}{3} - \epsilon & \frac{1}{2} & \frac{1-\epsilon}{2} \\ \frac{1-\epsilon}{2} & \frac{1+\epsilon}{2} & \frac{1}{2} \end{bmatrix}. \tag{25}$$

$\Phi^{\mathrm{ML}}$ yields a stochastic policy that depends on the value of $\epsilon$. For example, if $\epsilon = 1/12$, the resulting policy is $\pi = \left[\frac{1}{4}, \frac{1}{4}, \frac{1}{2}\right]$. However, as $\epsilon$ approaches 0, the resulting policy converges to $[0, 0, 1]$. This shows that $\pi'(y_3) \to 1$ while $w_3^\sigma = 1/3$, thus demonstrating that there exists no $\gamma_1 > 0$ for which $\Phi^{\mathrm{ML}}$ satisfies $\gamma$-PBM.

To show that $\Phi^{\mathrm{MB}}$ violates $\gamma$-PBM, consider the case with $M = 3$ where the profile $\sigma$ consists of the following three groups of evaluators:

$$\sigma = \{(y_1 \succ y_2 \succ y_3) \times 0.30, \ (y_2 \succ y_1 \succ y_3) \times 0.45, \ (y_3 \succ y_1 \succ y_2) \times 0.25\}, \tag{26}$$

where $(y_1 \succ y_2 \succ y_3)$ represents a ranking $r$ and "$\times 0.30$" indicates that $\sigma_r = 0.30$. Then, the Borda scores are calculated as $B = [1.3, 1.20, 0.5]$. Thus, $\Phi^{\mathrm{MB}}(\sigma) = \pi$, where $\pi(y_1) = 1$. Next, suppose the second group strategically misreports their preference from $(y_2 \succ y_1 \succ y_3)$ to $(y_2 \succ y_3 \succ y_1)$. Then, the Borda scores are calculated as $B' = [0.85, 1.2, 0.95]$. The resulting policy is then $\pi'(y_2) = 1$, with the population share of the second group being $w_2^\sigma = 0.45$. This example demonstrates that there exists no $\gamma_1 > 0$ for which $\Phi^{\mathrm{MB}}$ satisfies $\gamma$-PBM.

Next, we show that $\Phi^{\mathrm{RD}}$ satisfies all four axioms. $\Phi^{\mathrm{RD}}$ satisfies monotonicity because improving ranking of $y$ cannot decrease the number of evaluators whose top choice is $y$. In addition, $\Phi^{\mathrm{RD}}$ satisfies Pareto efficiency because if $r(y_j) < r(y_k)$ for all $r \in \mathcal{S}$ with $\sigma_r > 0$, then we have $w_k^\sigma = 0$ and $\Phi^{\mathrm{RD}}(\sigma)(y_k) = 0$. Additionally, $\Phi^{\mathrm{RD}}$ satisfies $\alpha$-PPA with $\alpha(\sigma) = 1$ for all $\sigma$ by its definition, and also satisfy $\gamma$-PBM with $(\gamma_1, \gamma_2) = (1, 0)$ because each group $G_k$ cannot increase $w_k^\sigma$ by manipulation.

## F  PROOF OF THE NON-IMPLEMENTABILITY OF $\Phi^{\mathrm{RD}}$

Suppose that $\Phi^{\mathrm{RD}}$ can be implemented by a preference learning algorithm $F^{\mathrm{RD}}$. Let $M = 3$, and consider two preference profiles, $\sigma_1$ and $\sigma_2$, defined as follows:

$$\begin{aligned} \sigma_1 &= \{(y_1 \succ y_2 \succ y_3) \times 1/3, \ (y_2 \succ y_1 \succ y_3) \times 1/3, \ (y_3 \succ y_1 \succ y_2) \times 1/3\}, \\ \sigma_2 &= \{(y_1 \succ y_2 \succ y_3) \times 2/3, \ (y_3 \succ y_2 \succ y_1) \times 1/3\}. \end{aligned} \tag{27}$$

Both of these profiles induce the same aggregated preference function $\mathrm{P}^\sigma = \mathrm{P}^{\sigma_1} = \mathrm{P}^{\sigma_2}$, where

$$\mathrm{P}^\sigma = \begin{bmatrix} \frac{1}{2} & \frac{2}{3} & \frac{2}{3} \\ \frac{1}{3} & \frac{1}{2} & \frac{2}{3} \\ \frac{1}{3} & \frac{1}{3} & \frac{1}{2} \end{bmatrix}. \tag{28}$$

Therefore, the preference learning algorithm $F^{\mathrm{RD}}$ would produce the same policy for both $\sigma_1$ and $\sigma_2$. However, according to the definition of $\Phi^{\mathrm{RD}}$, we have $\Phi^{\mathrm{RD}}(\sigma_1) = [1/3, 1/3, 1/3]$ and $\Phi^{\mathrm{RD}}(\sigma_2) = [2/3, 0, 1/3]$, which are different policies. This implies that $F^{\mathrm{RD}}$ does not implement $\Phi^{\mathrm{RD}}$, which contradicts our initial assumption. Therefore, $\Phi^{\mathrm{RD}}$ is not implementable by a preference learning algorithm.

## G    PROOF OF PROPOSITION 4.2

First, consider any feasible population share $w$ given a preference function P. By Definition 4.1, there exists a profile $\sigma$ such that $w = w^\sigma$ and $P = P^\sigma$. Then, the group-specific preference functions $(P_1^\sigma, \ldots, P_M^\sigma)$ that constitute $P^\sigma$, satisfy the condition in equation 3, which implies that $w \in \mathcal{W}(P)$.

Next, consider any $w \in \mathcal{W}(P)$. By the definition of $\mathcal{W}(P)$, there exist $(P_1, \ldots, P_M) \in \mathcal{P}^M$ such that $P = \sum_{k=1}^M w_k P_k$, where each $P_k$ satisfies $P_k(y_k \succ y) = 1$ for all $y \neq y_k$. Since $P_k \in \mathcal{P}$, there exists a profile $\sigma_k$ that induces $P_k$, such that $P_k = P^{\sigma_k}$. Now, if we consider an aggregated profile $\sigma := \sum_{k=1}^M w_k \sigma_k$ by combining these group profiles with the corresponding weights, then the preference function of $\sigma$ will be $P^\sigma = \sum_{k=1}^M w_k P^{\sigma_k} = \sum_{k=1}^M w_k P_k = P$ and also $w^\sigma = w$. Therefore, $w$ is a feasible population distribution given P.

## H    PROOF OF THEOREM 4.4

Consider any $w \in \mathcal{W}(P)$. Then, there exists $(P_1, \ldots, P_M) \in \mathcal{P}^M$ such that $P = \sum_{k=1}^M w_k P_k$. Fix an index $i \in [M]$. For any $y \in \mathcal{Y} \setminus \{y_i\}$, we have

$$P(y_i \succ y) = w_i P_i(y_i \succ y) + \sum_{k \neq i} w_k P_k(y_i \succ y) \geq w_i \tag{29}$$

since $P_i(y_i \succ y) = 1$. Taking the minimum over $y \in \mathcal{Y} \setminus \{y_i\}$ yields

$$\min_{y \in \mathcal{Y} \setminus \{y_i\}} P(y_i \succ y) \geq w_i, \tag{30}$$

which implies $w_i \leq u_i \ \forall i \in [M]$ and $w \in \overline{\mathcal{W}}(P)$. Therefore, $\mathcal{W}(P) \subseteq \overline{\mathcal{W}}(P)$.

## I    ADDITIONAL REMARKS ON THE TIGHTNESS OF THE OUTER APPROXIMATION

The gap between the true feasible set $\mathcal{W}(P)$ and its outer approximation $\overline{\mathcal{W}}(P)$ arises from our profile assumption, namely that each evaluator has a strict and complete ranking. To illustrate this point, we show that $\overline{\mathcal{W}}(P)$ provides a tight approximation (i.e., $\mathcal{W}(P) = \overline{\mathcal{W}}(P)$) under an extended profile setting. Consider an extended profile setting in which each group $G_k$ is allowed to provide pairwise comparison data according to its own preference function $P_k$, subject only to the skew-symmetry constraint $P_k(y_i \succ y_j) + P_k(y_j \succ y_i) = 1$ for all $y_i, y_j \in \mathcal{Y}$, and the unanimity constraint $P_k(y_k \succ y) = 1$ for all $y \in \mathcal{Y}$. In this case, the set $\mathcal{P}$ is defined as

$$\mathcal{P} := \{P \mid P(y \succ y') + P(y' \succ y) = 1 \ \forall y, y' \in \mathcal{Y}\}. \tag{31}$$

We show that $\overline{\mathcal{W}}(P) \subseteq \mathcal{W}(P)$ also holds under this setting. Consider any $w \in \overline{\mathcal{W}}(P)$. By assumption, $w$ satisfies $w_i \leq u_i = \min_{y \in \mathcal{Y} \setminus \{y_i\}} P(y_i \succ y)$ for all $i \in [M]$. Define each element of $(P_1, \ldots, P_M)$ as

$$P_k(y_i \succ y_j) = \frac{P(y_i \succ y_j) - w_i}{1 - w_i - w_j} \tag{32}$$

for any $i, j \neq k$, and let $P_k(y_k \succ y) = 1$, $P_k(y \succ y_k) = 0$ for all $y \in \mathcal{Y}$. Then $P_k(y_i \succ y_j) \in [0, 1]$ holds because $P(y_i \succ y_j) \in [w_i, 1 - w_j]$ by assumption. The skew-symmetry condition $P_k(y_i \succ y_j) + P_k(y_j \succ y_i) = 1$ is also satisfied. Thus, $P_k \in \mathcal{P}$ and $P_k$ can be induced by some profile. Finally, the constraint $P = \sum_{k=1}^M w_k P_k$ also holds. Therefore, $w \in \mathcal{W}(P)$, implying $\overline{\mathcal{W}}(P) \subseteq \mathcal{W}(P)$, and hence $\mathcal{W}(P) = \overline{\mathcal{W}}(P)$.

## J    CONNECTION OF THEOREM 4.4 AND TATLI ET AL. (2024)

Tatli et al. (2024) studies the recovery of population preference distributions under a spatial model. In their framework, each alternative is represented by a feature vector in a Euclidean space, and each voter's preferences are determined by distances to these vectors (i.e., voters prefer alternatives that are

closer in the Euclidean norm). Theorem 4.4 can also be derived in this setting. Specifically, consider a sufficiently high-dimensional feature space partitioned into $M!$ regions by $\binom{M}{2}$ hyperplanes, where each hyperplane is the perpendicular bisector of the line segment connecting a pair of alternative vectors. Then, (Tatli et al., 2024, Proposition 2) shows that it is impossible to recover the full profile $\sigma$ from aggregated pairwise comparison data $P^\sigma$. Moreover, by summing the inequality in (Tatli et al., 2024, Proposition 4) over the regions corresponding to voters who most prefer each alternative $y_i$, we obtain the bound $w_i \leq u_i$.

## K  IMPOSSIBILITY OF A UNIFORM GUARANTEE $\alpha(\sigma) > 2/M$

**Proposition K.1.** *No PSCF can be implemented by a preference–learning algorithm while guaranteeing $\alpha$-PPA with a constant $\alpha(\sigma) > 2/M$ for all $\sigma \in \Delta(\mathcal{S})$.*

*Proof.* Consider any setting in which the pairwise comparison data are completely uninformative. Specifically, suppose that for every pair of alternatives $y_i, y_j$, the observed probability satisfies $P(y_i \succ y_j) = 0.5$. Under such maximally ambiguous data, no preference–learning algorithm can distinguish among alternatives, and any algorithm that aims to maximize the worst-case $\alpha(\sigma)$ must output the uniform distribution over the $M$ alternatives.

However, the true distribution of evaluators' top choices may in fact be highly non-uniform while remaining perfectly consistent with the uninformative pairwise data. For example, consider a profile $\sigma$ in which $w^\sigma = [1/2, 1/(2M-2), 1/(2M-2), \ldots, 1/(2M-2)]$, corresponding to a situation where $y_1$ is ranked first by half of the evaluators and ranked last by the other half. In this case, the corresponding proportionality guarantee is $\alpha(\sigma) = 2/M$. Therefore, achieving a uniform lower bound $\alpha(\sigma) > 2/M$ for all $\sigma \in \Delta(\mathcal{S})$ is impossible. $\qquad\square$

## L  PROOF OF THEOREM 4.6

We first prove monotonicity. Improving the ranking of $y_i$ for some evaluator can only increase $P^\sigma(y_i \succ y)$ for any $y \neq y_i$, and decrease $P^\sigma(y \succ y_i)$. This implies that $u_i$ cannot decrease, while $u_j$ for $j \neq i$ cannot increase. Therefore, $\pi(y_i) = u_i/(\sum_{j=1}^M u_j)$ cannot decrease, establishing monotonicity.

Next, we prove Pareto efficiency. Suppose $y_i$ is ranked above $y_j$ in every input ranking, i.e., $r(y_i) < r(y_j)$ for all $r \in \mathcal{S}$ with $\sigma_r > 0$. Then, we have $P^\sigma(y_i \succ y_j) = \sum_{r \in \mathcal{S}} \sigma_r \cdot \mathbf{1}_{\{r(y_i) < r(y_j)\}} = 1$ and also $P^\sigma(y_j \succ y_i) = 0$. Thus, we get $u_j = \min_{y \in Y \setminus \{y_j\}} P^\sigma(y_j \succ y) = 0$. Thus, the resulting policy satisfies $\Phi^*(\sigma)(y_j) = u_j/(\sum_{k=1}^M u_k) = 0$. Therefore, $\Phi^*(\sigma)(y_i) \geq \Phi^*(\sigma)(y_j)$, establishing that $\Phi^*$ satisfies Pareto efficiency.

## M  PROOF OF LEMMA 4.7 AND LEMMA 4.9

Lemma 4.7 follows directly from the fact that $w_i^\sigma \leq u_i$ for all $i \in [M]$, which gives

$$\frac{\pi(y_i)}{w_i^\sigma} = \frac{u_i}{w_i^\sigma \sum_{j=1}^M u_j} \geq \frac{1}{\sum_{j=1}^M u_j}. \tag{33}$$

Next, we show Lemma 4.9. Let $I \subseteq [M]$ be the set of indexes for $\delta$-dominated alternatives, where $|I| = M - N_\delta^\sigma$. Then, for any $i \in I$, we have

$$u_i = \min_{j \in [M] \setminus \{i\}} P^\sigma(y_i \succ y_j) \leq P^\sigma(y_i \succ y_i') \leq 1 - \delta, \tag{34}$$

where $y_i'$ denotes an alternative that $\delta$-dominates $y_i$. Additionally, let $k \in \arg\max_{i \in [M]} w_i^\sigma$. Then, for any $i \neq k$, we have

$$u_i = \min_{j \in [M] \setminus \{i\}} P^\sigma(y_i \succ y_j) \leq P^\sigma(y_i \succ y_k) \leq 1 - w_k^\sigma. \tag{35}$$

Similarly, let $l \in \arg\max_{i \in [M], i \neq k} w_i^{\sigma}$, then we have $u_k \leq 1 - w_l^{\sigma}$. Combining these results,

$$\sum_{i=1}^{M} u_i = u_k + \sum_{i \neq k, i \notin I} u_i + \sum_{i \in I} u_i \leq (1 - w_l^{\sigma}) + (N_\delta^{\sigma} - 1)(1 - w_k^{\sigma}) + (M - N_\delta^{\sigma})(1 - \delta). \quad (36)$$

In addition, since $u_i \geq w_i^{\sigma}$, we have $\sum_{i=1}^{M} u_i \geq \sum_{i=1}^{M} w_i^{\sigma} = 1$. Combining both inequalities and plugging $(w_k^{\sigma}, w_l^{\sigma}) = (w^{\sigma,1}, w^{\sigma,2})$ in, we get the result of Lemma 4.9 as follows:

$$\left( \sum_{i=1}^{M} u_i \right)^{-1} \in \left[ \frac{1}{(N_\delta^{\sigma} - 1)(1 - w^{\sigma,1}) + (1 - w^{\sigma,2}) + (M - N_\delta^{\sigma})(1 - \delta)}, 1 \right]. \quad (37)$$

## N    ADDITIONAL BOUND ANALYSIS UNDER A RANDOM-RANKING MODEL

In this appendix, we provide additional analysis of the lower bounds in Lemma 4.7 and Lemma 4.9 under a random-ranking model. Our objective is to evaluate whether these guarantees remain meaningful under realistic levels of heterogeneity in the underlying ranking distributions.

**Setup.**    We construct a base ranking by sampling latent rewards for each of the $M$ alternatives independently from $\mathcal{N}(0, 1)$. The rankings of $N = 1000$ evaluators are then generated by adding independent Gaussian noise from $\mathcal{N}(0, 1)$ to these rewards and sorting the alternatives according to the perturbed values. This procedure induces a ranking distribution with substantial dispersion. For instance, when $M = 20$, the largest population share $\max_{k \in [20]} w_k^{\sigma}$ averages approximately $0.25$, indicating that the distribution is far from concentrated on a single ranking.

**Results.**    Table 3 reports the average values of the two lower bounds, $(\sum_{i=1}^{M} u_i)^{-1}$ and $\alpha(\sigma)$, over 10 independent runs for different values of $M$, with $\delta = 0.7$ fixed. In all cases, both bounds are substantially stronger than the naive baseline of $1/M$, corresponding to the uniform policy. These findings suggest that the theoretical guarantees remain nontrivial in empirically relevant regimes.

Table 3: Average lower bounds under the random-ranking model.

| $M$ | 10 | 20 | 50 | 100 |
|---|---|---|---|---|
| $\left( \sum_{i=1}^{M} u_i \right)^{-1}$ | 0.5553 | 0.3360 | 0.2085 | 0.1427 |
| $\alpha(\sigma)$ | 0.2539 | 0.1254 | 0.0570 | 0.0305 |

## O    PROOF OF THEOREM 4.11

Let $\sigma'$ be the profile manipulated by group $G_k$, and let $\pi' = \Phi^*(\sigma')$ be the resulting policy. $G_k$ aims to maximize

$$\pi'(y_k) = \frac{u_k'}{u_k' + \sum_{i \neq k} u_i'}, \quad (38)$$

where $u'$ represents the value of $u$ after the manipulation. To maximize $\pi'(y_k)$, $G_k$ will attempt to maximize $u_k'$ and minimize $\sum_{i \neq k} u_i'$. Since increasing the ranking of $y_k$ in their profile increases (or at least does not decrease) the value of $u_k'$ without increasing the value of $\sum_{i \neq k} u_i'$, the optimal strategy for $G_k$ is to truthfully report $y_k$ as its top choice. In this strategy, we have $u_k' = u_k$ and the sum $\sum_{i \neq k} u_i'$ has the following lower bound:

$$\sum_{i \neq k} u_i' \geq \sum_{i \neq k} w_i^{\sigma} = 1 - w_k^{\sigma}. \quad (39)$$

Substituting this lower bound into equation 38, we obtain

$$\pi'(y_k) \leq \frac{u_k}{u_k + 1 - w_k^{\sigma}} \leq \frac{1}{2}(w_k^{\sigma} + 1), \quad (40)$$

where the final inequality holds if $(w_k^{\sigma} - u_k + 1)(w_k^{\sigma} - 1) \leq 0$, which follows from the fact that $w_k^{\sigma}, u_k \in [0, 1]$.

## P    WEAK STRATEGYPROOFNESS GUARANTEE

In social choice theory, a mechanism is considered strategyproof if participants cannot benefit (i.e., increase their utility) by misreporting their true preferences (Gibbard, 1973), regardless of what other participants report. In our preference learning framework, we assume each group $G_k$'s utility is the probability assigned to its top choice, represented by $\pi(y_k)$. A preference learning algorithm is strategyproof if no participant can improve its outcome by misreporting preferences. However, as noted by Buening et al. (2025), strict strategyproofness is typically too restrictive and is not satisfied by the conventional preference learning algorithms (with ex-post efficiency). Our method does not satisfy strict strategyproofness like other methods, but satisfies a weaker form that provides a bounded guarantee on the maximum potential gain from strategic misreporting in equilibrium.

Let $\sigma'$ denote the profile resulting from strategical misreporting by all groups, and let $\pi' = \Phi^*(\sigma')$ be the resulting policy. Each group $G_k$ aims to maximize $\pi'(y_k)$, which involves maximizing $u'_k$ and minimizing $\sum_{i \neq k} u'_i$.

Since improving the ranking of $y_k$ in their reported preferences increases (or at worst, does not decrease) the value of $u'_k$ without increasing $\sum_{i \neq k} u'_i$, the optimal strategy for $G_k$ is to truthfully report $y_k$ as their top choice. Hence, all groups truthfully report their top choice regardless of other groups' strategies, meaning $P'_k(y_k \succ y) = 1$ for all $y \neq y_k$, where $P'_k$ denotes the reported preference function of $G_k$.

In this equilibrium, following steps analogous to the proof of Theorem 4.11, we have:

$$\pi'(y_k) \leq \frac{u'_k}{u'_k + 1 - w^\sigma_k} \leq \frac{1}{2}(w^\sigma_k + 1) \quad \forall k \in [M]. \tag{41}$$

Note that $\gamma(w^\sigma_k)$ is not a tight bound. Further exploration into tighter bounds and detailed analysis of each group's strategic behavior is left for future research.

## Q    PROOF OF PROPOSITION 4.13

Suppose $M = 2$ and $P^\sigma(y_1 \succ y_2) \in (0.5, 1)$, so $y_1$ is the Condorcet winner. If a PSCF $\Phi$ satisfies Condorcet consistency, it must return the deterministic policy $\pi(y_1) = 1$. However, this violates the $\alpha$-PPA axiom because $\pi(y_2) = 0$ while $w^\sigma_2 > 0$, which implies that $\pi(y_2)/w^\sigma_2 = 0$ cannot be lower bounded by any $\alpha(\sigma) > 0$.

## R    PROOF OF PROPOSITION 4.14

Suppose $y_i$ is a Condorcet winner. Then $P^\sigma(y_i \succ y_j) > 0.5$ for all $j \neq i$, which implies that $u_i > 0.5$. For any other $j \neq i$, we have $u_j \leq P^\sigma(y_j \succ y_i) < 0.5$. Therefore, $y_i$ has the highest $u_i$, i.e., $i \in \arg\max_{j \in [M]} u_j$, and $\Phi^\infty$ returns $\pi(y_i) = 1$, satisfying Condorcet consistency.

## S    FINITE BEHAVIOR OF $\Phi^\beta$

The following proposition quantifies how large the parameter $\beta$ needs to be to ensure that a Condorcet winner receives a sufficiently high probability under the softmax policy.

**Proposition S.1** (Condorcet consistency at finite $\beta$). *Let $y_i$ be a Condorcet winner with $u_i > 0.5$. Then, the softmax policy satisfies $\pi(y_i) \geq \alpha_c$ if*

$$\beta \geq \frac{1}{u_i - 0.5} \log\left(\frac{(M-1)\alpha_c}{2(1 - \alpha_c)}\right). \tag{42}$$

*Proof.* Since $y_i$ is a Condorcet winner, we have $u_j \leq P^\sigma(y_j \succ y_i) = 1 - P^\sigma(y_i \succ y_j) < 0.5$ for any $j \neq i$. From the given condition

$$\beta \geq \frac{1}{u_i - 0.5} \log\left(\frac{(M-1)\alpha_c}{2(1 - \alpha_c)}\right), \tag{43}$$

we can establish the following lower bound:

$$u_i \exp\left(\beta u_i\right) \geq \frac{\alpha_c}{1 - \alpha_c} (M - 1)(0.5 \exp\left(0.5\beta\right)) \geq \frac{\alpha_c}{1 - \alpha_c} \sum_{j \neq i} u_j \exp\left(\beta u_j\right). \qquad (44)$$

Thus, the softmax policy satisfies

$$\pi(y_i) = \frac{u_i \exp\left(\beta u_i\right)}{u_i \exp\left(\beta u_i\right) + \sum_{j \neq i} u_j \exp\left(\beta u_j\right)} \geq \alpha_c. \qquad (45)$$

$\square$

In addition, it can be shown that the $\alpha$-PPA guarantee deteriorates as $\beta \to \infty$, since the lower bound in Lemma 4.7 becomes

$$\frac{\pi(y_i)}{w_i^\sigma} \geq \left( \sum_{j=1}^{M} u_j \exp\left(\beta(u_j - u_i)\right) \right)^{-1}, \qquad (46)$$

which converges to zero as $\beta \to \infty$, unless $u_i = \max_{j \in [M]} u_j$.

## T  CONNECTION TO PAIRWISE MAJORITY CONSISTENCY (PMC)

We discuss the connection to pairwise majority consistency (PMC) (Ge et al., 2024), which imposes a stronger consistency requirement, ensuring the entire policy ranking aligns with majority preferences.

**Definition T.1** (Pairwise majority consistent ranking (PMC ranking)). A ranking $r^\sigma$ is a called a *PMC ranking* of a profile $\sigma$ if for all $y_i, y_j \in \mathcal{Y}$, a majority of evaluators prefer alternative $y_i$ to alternative $y_j$ in $\sigma$ if and only if $y_i$ is ranked higher than $y_j$ in $r^\sigma$. Formally, $\mathrm{P}^\sigma(y_i \succ y_j) > 1/2$ if and only if $r^\sigma(y_i) < r^\sigma(y_j)$.

**Definition T.2** (Pairwise majority consistency (PMC)). A PSCF $\Phi$ satisfies PMC if, for any profile $\sigma \in \Delta(\mathcal{S})$ that has a PMC ranking $r^\sigma$, $\Phi(\sigma)$ has the same ranking with $r^\sigma$, i.e. $\Phi(\sigma)(y_i) \geq \Phi(\sigma)(y_j)$ if $r^\sigma(y_i) < r^\sigma(y_j)$.

It can be shown that any $\Phi^\beta$ with finite $\beta \geq 0$ violates PMC, and only the limiting PSCF $\Phi^\beta$ satisfies PMC.

**Proposition T.3.** *Any $\Phi^\beta$ with finite $\beta \geq 0$ violates PMC. $\Phi^\infty$ satisfies PMC.*

*Proof.* First, we show that $\Phi^\beta$ violates PMC for any $\beta \geq 0$. It suffices to demonstrate that there exists a profile $\sigma$ with a PMC ranking $r^\sigma$ for which $\Phi^\beta(\sigma)(y_i) < \Phi^\beta(\sigma)(y_j)$ while $r^\sigma(y_i) < r^\sigma(y_j)$.

Consider the following profile $\sigma$ with $M = 3$:

$$\sigma = \{(y_1 \succ y_2 \succ y_3) \times 0.3,\ (y_2 \succ y_3 \succ y_1) \times 0.1,\ (y_3 \succ y_1 \succ y_2) \times 0.3,\ (y_3 \succ y_2 \succ y_1) \times 0.3\}, \qquad (47)$$

which yields the following preference function:

$$\mathrm{P}^\sigma = \begin{bmatrix} 0.5 & 0.6 & 0.3 \\ 0.4 & 0.5 & 0.4 \\ 0.7 & 0.6 & 0.5 \end{bmatrix}. \qquad (48)$$

Then, the PMC ranking satisfies $r^\sigma(y_3) < r^\sigma(y_1) < r^\sigma(y_2)$, as $\mathrm{P}^\sigma(y_3 \succ y_1), \mathrm{P}^\sigma(y_3 \succ y_2), \mathrm{P}^\sigma(y_1 \succ y_2) > 0.5$. However, we have $u_1 = 0.3$ and $u_2 = 0.4$. Since $u_1 < u_2$, it follows that $\Phi^\beta(\sigma)(y_1) < \Phi^\beta(\sigma)(y_2)$ regardless of $\beta$, contradicting $r^\sigma(y_1) < r^\sigma(y_2)$. Therefore, $\Phi^\beta$ violates PMC regardless of $\beta$.

Next, we show that $\Phi^\infty$ satisfies PMC. Consider a profile $\sigma$ with its PMC ranking $r^\sigma$, and let $y^* \in \mathcal{Y}$ be the alternative ranked first in $r^\sigma$ (i.e., $r^\sigma(y^*) = 1$). Then, $y^*$ must be a Condorcet winner, as $\mathrm{P}^\sigma(y^* \succ y) > 1/2$ for all $y \neq y^*$. Thus, $\pi := \Phi^\infty(\sigma)$ is a deterministic policy with $\pi(y^*) = 1$. Consequently, we have $\pi(y^*) > \pi(y)$ for all $y \neq y^*$ and trivially $\pi(y_i) = \pi(y_j) = 0$ for any $y_i, y_j \neq y^*$, satisfying the condition for PMC. $\square$

$\Phi^\beta$ approximately satisfies PMC as $\beta \to \infty$ if we allow some slack in the rankings (e.g., $\Phi(\sigma)(y_i) \geq \Phi(\sigma)(y_j) - \epsilon$ for some small $\epsilon > 0$) in the definition of PMC. However, exploring this approximate consistency is beyond the scope of this paper and is left for future research.

# U    ADDITIONAL DETAILS OF EXPERIMENTS

## U.1    SYNTHETIC DATASET

**Dataset generation.**    For the synthetic dataset, we used 10 prompts and 10 responses for the color-preference alignment task, as shown in Table 4. To construct the ground-truth profile $\sigma$, we sampled the true (center) rewards independently from the normal distribution $\mathcal{N}(0,1)$ for each response. We then added i.i.d. random noise from $\mathcal{N}(0,1)$ to each true reward to generate $1{,}000$ independent rankings. Finally, we drew $10^4$ pairwise comparison samples i.i.d. from the true preference function $P^\sigma$ to train each algorithm.

Table 4: Prompts and responses in synthetic dataset

| Prompt ($x$) | Response ($y$) |
| --- | --- |
| Which color do you find the most appealing? | Red |
| Which color best represents your personality? | Blue |
| When decorating your room, what color do you prefer? | Green |
| What is your favorite color? | Yellow |
| Which color do you like the most? | Purple |
| If you had to choose just one color, which would it be? | Orange |
| Among all colors, what's your top pick? | Pink |
| If you could only wear one color forever, what would you choose? | Brown |
| What color makes you feel happiest? | Black |
| Which color do you prefer most? | White |

**Evaluation methods.**    We evaluate the fine-tuned policy using two metrics: (i) win rate against a reference policy (the pretrained model), $\mathbb{E}_{(x,y_1,y_2)\sim(\rho,\pi,\pi_{\mathrm{ref}})}[P^\sigma(y_1 \succ y_2 \mid x)]$, and (ii) the PPA level $\alpha(\sigma)$. To estimate the fine-tuned policies over responses, we compute the logits of each response and the softmax policy (with temperature 1). We then calculate the win rate and PPA level directly from their definitions using the estimated policy for each prompt. The results are averaged over all prompts.

## U.2    ALPACA-GPT4 DATASET

**Dataset generation.**    We considered two groups of evaluators, defined across two categories: expertise and style. For the expertise category, evaluators were grouped into two levels: 'elementary school student' and 'PhD student'. For the style category, evaluators were grouped into 'friendly' and 'unfriendly'. The true population distribution was set to $w^\sigma = [0.8, 0.2]$. For each of the 52k instruction prompts from the Alpaca-GPT4 dataset (Peng et al., 2023), group-specific responses were generated using GPT-4.1 with the prompts listed in Table 5. Then, the pairwise comparison samples are drawn i.i.d. from $P^\sigma$.

Table 5: Prompts used for generating responses from each group

| Category | Prompt |
| --- | --- |
| Expertise | (1) Generate a response that can be easily understood by an elementary school student. 
 (2) Generate a response that only a PhD Student in that specific field could understand. |
| Style | (1) Generate a response that is friendly, witty, funny, and humorous, like a close friend. 
 (2) Generate a response that answers in an unfriendly manner. |

**Evaluation methods.**    To estimate the fine-tuned policies over responses, we sample a response from the policy and use the annotation model (GPT-4.1) to classify its group. Table 6 shows the

prompts used to classify the group of generated responses. Based on these classifications, we evaluate the policy's win rate and the PPA level from their definitions.

Table 6: Prompts used for classification

| Category | Prompt |
|---|---|
| Expertise | Does the expertise level of this response align more closely with the elementary level or the PhD student level? Please answer with only one of these exact options: 'elementary' or 'PhD'. |
| Style | Is this response friendly or unfriendly? Please answer with only one of these exact options: 'friendly' or 'unfriendly'. |

### U.3 HYPERPARAMETER SETTING

The Qwen2.5-3B-Instruct model (Yang et al., 2024) was fine-tuned using each algorithm, where both the reference policy $\pi_{\text{ref}}$ and the data sampling policy $\pi_d$ were set to the same pretrained model. All algorithms were trained on the same offline dataset for the same number of iterations. NLHF was not included in the comparison, as the algorithm does not support offline learning. Specific training hyperparameters are provided in Table 7. Each training run utilized one H100 GPU, requiring approximately 0.5–1 hour per epoch with about 20–40GB of memory usage using LoRA.

Table 7: Training hyperparameters

| Hyperparameter | Synthetic | Alpaca-GPT4 |
|---|---|---|
| Training & Reference Model | Qwen2.5-3B-Instruct | Qwen2.5-3B-Instruct |
| Learning Rate | 1e-4 | 1e-5 |
| Batch Size | 8 | 4 |
| Epochs | 3 | 1 |
| Optimizer | AdamW | AdamW |
| Gradient Clipping | 1.0 | 1.0 |
| Learning Rate Scheduler | Linear | Linear |
| Warmup Steps | 100 | 100 |
| KL Coefficient | 0.1 | 0.01 |
| LoRA Rank | 32 | 32 |
| LoRA $\alpha$ | 32 | 32 |

## V   SCALABLE OFFLINE ALGORITHM WITH FUNCTION APPROXIMATION

### V.1   OFFLINE PAIRWISE COMPARISON DATASET

In practical applications of preference learning, the preference function often depends on additional context or state, denoted by $x \in \mathcal{X}$. For instance, in LLMs, $x$ represents the input prompt or conversational history that provides the specific context for generating a preferred response. Accordingly, we define the context-dependent preference function as $\mathrm{P}(\cdot \succ \cdot \mid \cdot) : \mathcal{Y}^2 \times \mathcal{X} \to [0, 1]$, which is unknown and must be estimated from empirical data. We consider an offline dataset of pairwise comparisons $\mathcal{D} = \{(x_i, y_i^w, y_i^l)\}_{i=1}^N$, where $y_i^w$ is preferred over $y_i^l$ under context $x_i$. Each query is assumed to be drawn i.i.d. from a joint distribution of $\rho(x)$ and $\pi_d(y \mid x)$, and labeled according to the preference function $\mathrm{P}$. Our goal is to use this offline dataset to learn a policy $\pi : \mathcal{X} \mapsto \Delta(\mathcal{Y})$ following the framework introduced in the previous sections.

## V.2 Two-Phase offline preference learning algorithm

We approximate a softmax policy proposed in Section 4.3:

$$\pi(y \mid x) := \frac{u(y \mid x) \exp(\beta u(y \mid x))}{\sum_{y \in \mathcal{Y}} u(y \mid x) \exp(\beta u(y \mid x))}, \quad \text{where} \quad u(y \mid x) := \min_{z \in \mathcal{Y}} \mathrm{P}(y \succ z \mid x). \quad (49)$$

Specifically, we use a two-phase algorithm that first estimates $u$ and then estimates $\pi$ based on $u$.

**Phase 1: Estimating $u$.** To estimate $u$, we first train the selector model $\mu$ using the following loss function, the offline dataset $\mathcal{D}$, and the parameterized function class $\mathcal{F}_\mu$:

$$\hat{\mu} \in \underset{\mu \in \mathcal{F}_\mu}{\arg \min} \frac{1}{N} \sum_{i=1}^{N} \frac{\mu(y_i^l \mid x_i, y_i^w)}{\pi_d(y_i^l \mid x_i)}. \quad (50)$$

Then, the estimated $\hat{u}$ can be obtained from $\hat{u}(y \mid x) = \sum_{z \in \mathcal{Y}} \hat{\mathrm{P}}(y \succ z \mid x)\hat{\mu}(z \mid x, y)$, where $\hat{\mathrm{P}}$ denotes the empirical estimate of the preference function. The derivation of the loss function and the relationship between $\mu$ and $u$ are provided in Appendix W.

**Phase 2: Estimating $\pi$.** Let $\hat{\pi}_\beta$ be the normalized softmax policy constructed with $\hat{u}$ following equation 49. In the second phase, the policy model is trained by minimizing the distance to $\hat{\pi}_\beta$ over a function class $\mathcal{F}_\pi$:

$$\hat{\pi} \in \underset{\pi \in \mathcal{F}_\pi}{\arg \min} \mathbb{E}_{x \sim \rho} \Big[ L^\pi(\pi(\cdot \mid x), \hat{\pi}_\beta(\cdot \mid x)) \Big]. \quad (51)$$

Here, $L^\pi$ denotes a divergence or distance metric between two policies. In our experiments, we employ the KL divergence for $L^\pi$. Specifically, substituting the KL divergence into $L^\pi$ from equation 51, the loss function becomes

$$\mathbb{E}_{x \sim \rho} \left[ D_{\mathrm{KL}} \left( \pi(\cdot \mid x) \middle\| \sum_{z \in \mathcal{Y}} \hat{P}(\cdot \succ z \mid x)\hat{\mu}(z \mid x, \cdot) \right) \right]$$

$$= \mathbb{E}_{(x,y) \sim (\rho, \pi_d)} \left[ \frac{\pi(y \mid x)}{\pi_d(y \mid x)} \log \frac{\pi(y \mid x)}{\sum_{z \in \mathcal{Y}} \hat{P}(y \succ z \mid x)\hat{\mu}(z \mid x, y)} \right]. \quad (52)$$

Using the offline dataset and function approximation, we obtain

$$\hat{\pi} \in \underset{\pi \in \mathcal{F}_\pi}{\arg \min} \mathbb{E}_{(x,y^w,y^l) \sim D} \left[ \frac{\pi(y^w \mid x)}{\pi_d(y^w \mid x)} \log \frac{\pi(y^w \mid x)}{\left(\frac{1}{2} + \frac{1}{2}\hat{\mu}(y^w \mid x, y^l)\right) \exp\left(\frac{\beta}{2} + \frac{\beta}{2}\hat{\mu}(y^w \mid x, y^l)\right)} \right.$$

$$\left. + \frac{\pi(y^l \mid x)}{\pi_d(y^l \mid x)} \log \frac{\pi(y^l \mid x)}{\left(\frac{1}{2} - \frac{1}{2}\hat{\mu}(y^l \mid x, y^w)\right) \exp\left(\frac{\beta}{2} - \frac{\beta}{2}\hat{\mu}(y^w \mid x, y^l)\right)} \right]. \quad (53)$$

## V.3 Additional techniques for LLM fine-tuning

**Regularization via reference policy.** Fine-tuning large language models (LLMs) requires maintaining alignment with a reference policy, typically the pretrained model. To prevent excessive drift, we incorporate KL-divergence regularization terms into the training objectives for both the selector model $\mu$ and the policy model $\pi$. Specifically, we add the following regularization terms to the loss functions in Phase 1 and Phase 2:

$$\beta_\mu \mathbb{E}_{(x,y) \sim (\rho, \pi_d)} \left[ D_{\mathrm{KL}} \left( \mu(\cdot \mid x, y) \middle\| \pi_{\mathrm{ref}}(\cdot \mid x, y) \right) \right], \quad \beta_\pi \mathbb{E}_{x \sim \rho} \left[ D_{\mathrm{KL}} \left( \pi(\cdot \mid x) \middle\| \pi_{\mathrm{ref}}(\cdot \mid x) \right) \right] \quad (54)$$

**Training with single model.** To reduce computational cost, we propose to train both $\mu$ and $\pi$ using a single model. This is enabled by encoding structural differences through specialized input formats. Specifically, $\pi(\cdot \mid x)$ selects preferred responses given a prompt, while $\mu(\cdot \mid x, y)$ selects responses given a prompt and a candidate response. By distinguishing these cases with separator tokens, we achieve performance comparable to training separate models, while improving memory usage and training efficiency.

# W   DERIVATION OF THE LOSS FUNCTION IN PHASE 1

**Step 1: LP reformulation.**   Recall the definition $u(y \mid x) := \min_{z \in \mathcal{Y} \setminus \{y\}} \mathrm{P}(y \succ z \mid x)$. Each $u(y \mid x)$ can be rewritten as

$$u(y \mid x) = \sum_{z \in \mathcal{Y}} \mathrm{P}(y \succ z \mid x) \mu^*(z \mid x, y), \tag{55}$$

where a *selector distribution* $\mu^*(\,\cdot\, \mid x, y) \in \Delta(\mathcal{Y})$ places all its mass on the minimizer of $\mathrm{P}(y \succ \cdot \mid x)$. Such $\mu^*$ and the corresponding pointwise minimum can be obtained via the following linear programming (LP):

$$u(y \mid x) = \min_{\mu(\cdot \mid x, y) \in \Delta(\mathcal{Y})} \sum_{z \in \mathcal{Y}} \mathrm{P}(y \succ z \mid x) \mu(z \mid x, y). \tag{56}$$

**Step 2: Aggregation of pointwise LPs.**   Assume the data-generating distribution $\rho(\cdot)$ and $\pi_d(\,\cdot\, \mid x)$ have full support. Multiplying and dividing equation 56 by $\pi_d(z \mid x)$ and then taking the expectation over all $z \in \mathcal{Y}$ gives

$$u(y \mid x) = \min_{\mu(\cdot \mid x, y) \in \Delta(\mathcal{Y})} \mathbb{E}_{z \sim \pi_d(\cdot \mid x)} \left[ \frac{\mathrm{P}(y \succ z \mid x) \mu(z \mid x, y)}{\pi_d(z \mid x)} \right]. \tag{57}$$

Next, we aggregate these pointwise LPs by multiplying each pointwise objective by $\rho(x)\pi_d(y \mid x)$ and summing over all $(x, y) \in \mathcal{X} \times \mathcal{Y}$. We also add the symmetrical term with swapped $y$ and $z$, which does not change the optimal solution:

$$\mu^* \in \underset{\mu: \mathcal{X} \times \mathcal{Y} \mapsto \Delta(\mathcal{Y})}{\arg\min} \mathbb{E}_{(x, y, z) \sim (\rho, \pi_d, \pi_d)} \left[ \frac{\mathrm{P}(y \succ z \mid x) \mu(z \mid x, y)}{\pi_d(z \mid x)} + \frac{\mathrm{P}(z \succ y \mid x) \mu(y \mid x, z)}{\pi_d(y \mid x)} \right]. \tag{58}$$

**Step 3: Empirical counterpart.**   Given an offline preference dataset $\mathcal{D}$, we approximate the expectation in equation 58 using its empirical counterpart and restrict the function class to a parameterized family $\mathcal{F}_\mu$:

$$\hat{\mu} \in \underset{\mu \in \mathcal{F}_\mu}{\arg\min} \frac{1}{N} \sum_{i=1}^{N} \frac{\mu(y_i^l \mid x_i, y_i^w)}{\pi_d(y_i^l \mid x_i)}. \tag{59}$$

Given the estimate $\hat{\mu}$, we can estimate $\hat{u}$ using equation 56 with the estimated preference function $\hat{\mathrm{P}}$:

$$\hat{\mathrm{P}}(y \succ z \mid x) := \begin{cases} \frac{N(x, y, z)}{N(x, y, z) + N(x, z, y)} & \text{if } N(x, y, z) + N(x, z, y) > 0, \\ 1/2, & \text{otherwise,} \end{cases} \tag{60}$$

where $N(x, y, z) := |\{i \in [N] \mid (x_i, y_i^w, y_i^l) = (x, y, z)\}|$.

