# OpenReview forum: "Beyond RLHF and NLHF: Population-Proportional Alignment under an Axiomatic Framework"
_ICLR.cc/2026/Conference — ICLR 2026 Poster_

### Official Review · Reviewer_YHTf · 2025-10-22

**Soundness:** 4
**Presentation:** 3
**Contribution:** 3
**Rating:** 6
**Confidence:** 3

**Summary:**

This work proposes an approach to better represent human preferences from a diverse population, ie reflecting the mix of views not just the majority, in a trained policy. It comes with guarantees on its proportionality to the actual population (population‑proportional alignment, or PPA) and its manipulation by a group (population‑bounded manipulability, PBM). And with a parameter knob to trade off proportionality against picking the Condorcet winner (the one option that beats all the others, pairwise) Their experiments show that they are able to get a good PPA with a win rate on par with RLHF, offering no proportionality.

**Strengths:**

- RLHF/NLHF tend to pick a single preference even when the "camps" are close; their approach better handles mix views from a diverse population of evaluators and align a policy according to its distribution ;
- it offers axiomatic guarantees, first monotonicity (if an option pairwise standing improves its probability doesn't go down) and Pareto efficiency (if everyone prefers $y$ to $y’$, then $y$ gets at least as much weight as $y’$), and their proposed PPA (every option gets at least a fraction of its true population share) and PBM (the influence of a group is bounded by a function of its share) ;
- the $\beta$ parameter adds flexibility between converging to Condorcet (higher $\beta$) or aiming for proportionality (lower $\beta$) ;
- they confirm their predicted results (with RHLF, NHLF, and their algorithm), observing the  $\beta$-induced trade-off.

**Weaknesses:**

- this is not an easy read: the topic has inherent complexity but the intuition for the algorithm is not easy to grasp from the math-heavy notation —it also means that, under time constraint, I cannot guarantee that sections 3 and 4 are fully sound ;
- despite this machinery, the true share of the population remains elusive, and a conservative strategy has to be picked ;
- it would have been nice to better illustrate the theory with use cases clearly showing how the $\beta$ parameter can be adjusted in situation ;
- the second, large-scale, experiment, shows a muted trade‑off on Alpaca‑GPT4, so the central PPA claim is hard to appreciate —the suggestion that "this is because the group distributions of outputs are estimated using an annotation model" is not satisfying regarding that claim.

**Questions:**

- out of curiosity, how would the approach be adapted to handle more than pairwise comparisons, e.g. respect the ranking of multiple options by a diverse population ?
- the influence of a single group is bounded, but isn't there a risk of collusion amongst groups?

---

> ### Author Response · Authors · 2025-11-21
>
> We are grateful to the reviewer for their positive assessment and thoughtful suggestions. Our responses to each concern are provided below.
>
> &nbsp;
>
> **1. The topic has inherent complexity but the intuition for the algorithm is not easy to grasp from the math-heavy notation.**
>
> We appreciate the reviewer’s feedback regarding readability and the difficulty of parsing the intuition behind our algorithm, due to the math-heavy notation.
> We have attempted to use only the minimal mathematical notation necessary and to follow conventions widely used in the social choice and alignment literature, with all symbols clearly documented in Appendix A.
> In the revised draft, we will re-examine the notation to further simplify and streamline it wherever possible. In addition, we will expand the high-level overview at the beginning of Sections 3 and 4 to present the core ideas and design intuition before introducing the formal details.
>
> &nbsp;
>
> **2. The true share of the population remains elusive, and a conservative strategy has to be picked.**
>
> We appreciate the reviewer’s insightful feedback. We would like to clarify that recovering the exact population distribution is fundamentally impossible under our data setting, where group labels are unobserved and only pairwise-comparison data are available. This limitation is formalized in our impossibility result (see Appendix K), which shows that no algorithm can guarantee a constant $\alpha > 2/M$ under such minimal information.
> For this reason, our method intentionally adopts a conservative strategy based on the feasible set of population distributions.
> Recovering the true population distribution more accurately would require richer forms of feedback or additional annotations. While this is an important avenue for future research, it falls outside the scope of the present paper, whose aim is to establish the foundational results.
>
> &nbsp;
>
> **3. Use cases clearly showing how the $\beta$ parameter can be adjusted.**
>
> We appreciate the reviewer’s suggestion to better illustrate practical use cases for adjusting the parameter $\beta$. While the exact choice of $\beta$ ultimately reflects a governance decision, our framework provides a simple, evaluation-driven procedure for selecting and tuning it in practice.
>
> First, one can construct a small evaluation set that reflects the intended deployment domain, where group labels can be reliably inferred. In LLM settings, group labels can be approximated using simple keyword-based rules or by comparing logits across candidate responses, as demonstrated in the synthetic experiments in Section 5.2.
>
> Next, given a desired target range for the proportional alignment level $\alpha(\sigma)$, one can tune $\beta$ by starting from a conservative value (e.g., $\beta = 10^2$) and adjusting it upward or downward while monitoring the resulting PPA estimate on the evaluation set. Since $\alpha(\sigma)$ increases monotonically with decreasing $\beta$, this tuning process is straightforward. We have added this practical procedure to Section 5.2 in the revised draft.
>
> As a simple illustration, consider a news-summarization model deployed to two audience groups: a large general-audience group that prefers short, high-level summaries, and a smaller expert group that prefers longer, more technical summaries. In this setting, adjusting the parameter $\beta$ controls how strongly the model’s output reflects the majority preference. To tune $\beta$ in practice, one can first construct a small evaluation set where group labels (general vs. expert) can be reliably inferred. During fine-tuning, $\beta$ can then be adjusted by comparing the estimated proportional alignment level $\alpha$ on this evaluation set with the target $\alpha$ desired for the application.

---

> ### Author Response · Authors · 2025-11-21
>
> **4. The large-scale experiment, shows a muted trade‑off on Alpaca‑GPT4, so the central PPA claim is hard to appreciate.**
>
> As the reviewer pointed out, the results on the Alpaca–GPT4 dataset exhibit a weaker trade-off.
> We would like to emphasize, however, that this limitation arises from the **evaluation method**, not the framework or algorithm itself.
> As shown in Section 5.2, when group identities are directly observable as in the synthetic color dataset where group labels can be obtained from model logits, the method exhibits a clear and consistent trade-off across values of $\beta$. This confirms that the LLM function-approximation version of our algorithm behaves as theoretically predicted.
>
> The noisiness in the results on the Alpaca-GPT4 dataset stems from the dataset and group definitions themselves. Categories such as "expertise" and "style" are inherently ambiguous, and distinguishing these attributes is challenging even for human annotators. This ambiguity inflates the estimated PPA level and weakens the contrast across $\beta$, which can indeed be observed in our reported results. We intentionally followed prior work (Jang et al. 2023; Chakraborty et al. 2024) in using these categories to ensure comparability, but they are not ideal for evaluating proportional alignment.
>
> To address this concern, we will include an additional experiment on the Alpaca–GPT4 dataset using a more clearly distinguishable group attribute. Specifically, we define groups based on the presence of certain keywords in the model’s output, which yields more reliable labels with far less ambiguity or classifier noise.
> For example, an output is assigned to group 1 if it contains any word from a predefined list of positive-energy or confidence-related descriptors ("Absolutely", "Definitely", "Clearly", "Undoubtedly", "Surely", "Certainly", "Positively", "Confidently", "Yes", "Of course"), and to group 2 otherwise.
> Following the same training procedure as in Section 5.2, we fine-tuned Qwen2.5-3B-Instruct using 10k Alpaca-GPT4 preference samples and evaluated on 1k test samples. Preliminary single-episode results under $w^\sigma = [0.4, 0.6]$ are shown below:
>
> | Metric  | $\beta=0$  | $\beta=10^{-4}$  |  $\beta=10^{-2}$ | $\beta=10^0$  | DPO  |
> |---|---|---|---|---|---|
> | PPA ($\alpha$)  |  0.7375 | 0.5925  |  0.2750 | 0.3450  | 0.1200 |
> | Win rate  |  0.5010 |  0.5068 | 0.5195  | 0.5167  |  0.5257 |
>
> These results show a much more pronounced change than in our earlier Alpaca–GPT4 experiment, confirming that when group labels are reliably identifiable, the behavior of our algorithm becomes clearly observable and matches the theoretical predictions.
> We will include the full results (using multiple episodes) as well as this discussion in the revised draft.
>
> &nbsp;
>
> **5. How would our approach extend beyond pairwise comparisons?**
>
> We thank the reviewer for raising this interesting question.
> Our approach can be directly extended to the $k$-wise comparison setting, where the feasible set becomes smaller as $k$ increases.
> Specifically, if top choices (or rankings) over triplets (i.e., $k=3$) are observed so that we can estimate quantities such as $P(y_i \succ y_j, y_l)$, then the feasible set constraint naturally strengthens from $w_i \leq \min_j P(y_i \succ y_j)$ to $w_i \leq \min_{(j, l)} P(y_i \succ y_j, y_l)$.
> As $k$ grows and richer comparison information becomes available, this polytope shrinks; in the limit when full rankings over all $M$ candidates are observable, the feasible set collapses to a singleton representing the exact population distribution. We view a systematic quantification of how the feasible set contracts with increasing $k$ as an interesting direction for future work.
>
> &nbsp;
>
> **6. Risk of collusion among groups.**
>
> In our problem setting, the impact of collusion across groups is inherently limited. Individuals within each group share the same top choice, and a group’s utility corresponds to the probability assigned to that top choice in the output distribution. Because of this structure, collusion across groups may increase the probability of their preferred alternatives slightly, but such improvements are necessarily small. Importantly, the PBM bound established in Theorem 4.11 continues to hold under collusion, unless at least one group sacrifices its own utility.

---

> ### Comment · Reviewer_YHTf · 2025-11-26
>
> thank you for addressing what I saw as weaknesses, and answering my questions. Having read the other reviewers' comments as well, I believe my assessment is fair as is.

---

### Official Review · Reviewer_JTvB · 2025-10-27

**Soundness:** 3
**Presentation:** 2
**Contribution:** 2
**Rating:** 4
**Confidence:** 3

**Summary:**

The paper targets distributional pluralism without group labels. It introduces PPA (probability mass at least proportional to each group's top choice share) and PBM (single-group manipulation is bounded by its true share) and gives a simple rule by considering the worst pairwise win it has against any other option and then set each option's probability to that worst win score, rescaled. There's a temperature knob shifting probabilities into the strongest options. The authors offer some experimental validation of this approach.

**Strengths:**

It clearly exposits what RLHF/NLHF does not guarentee. It provides theoretical analysis to back the proposed rule, and the rule itself is simple with a tunable $\beta$ term, and bridges social choice theory and preference learning.

**Weaknesses:**

There are a few issues I have with the paper.
- First of all, even though the author motivates with examples, the proposed method exhibits a clear trade-off as shown in the experiment: as beta increases, there's less pluralism but the performance increases. The paper shows the trade-off but offers little operational guidance for picking beta in real systems where “the population” is dynamic and contested. This pushes heavy governance choices onto practitioners.

- Perhaps most concerning to me is a lack of evidence that this method works. The only LLM study fine-tunes Qwen2.5-3B-Instruct; NLHF isn’t included, and group proportions in Alpaca are inferred by GPT-4.1 classifiers (no human labels). The authors themselves say differences across beta are "less distinct" due to annotator noise and that measuring PPA for LLMs are still an open challenge. This cast doubt on practical employability. The clearest empirical story is the tabular MovieLens task (1,297 rankings, 20 movies), where PPA/PBM can be computed exactly and the trade-off is clean; but this is not an LLM domain, so generalization to instruction tuning is unclear.

- PBM is weaker than strategy-proofness and difficult to measure in practice. Because it relies on unknown group labels, its guarantees aren’t verifiable. It limits—but does not eliminate—misreporting incentives; when manipulation is cheap, the remaining incentives can still be significant.

- As the authors pointed out in sec 4.3, the proposed rule is not condorcet consistent - even with a majority winner, it deliberately allocates mass to minority options, which comes at odds with desirable notions in many application which wants to pick the majority winner.

- The method only observes head-to-head outcomes and not true group sizes. It falls back on conservative upper bounds but when many matchups are close, the resulting policies become nearly flat even if one group is actually much larger.

**Questions:**

- Is there a principled way to choose or adapt beta across contexts and over time (under distribution shift), especially when the two desiderata are hard to measure reliably and given the trade-offs noted in the paper?

- Can the authors validate on more standard RLHF-style settings (preferably with human or auditor-verified cohort labels, etc.), given that the current Alpaca example relies on classifier-inferred groups and model-generated outputs and seems too noisy to support firm conclusions?

---

> ### Author Response · Authors · 2025-11-21
>
> We thank the reviewer for their thoughtful and insightful feedback. Our detailed responses to each comment are provided below. We hope these explanations will be helpful in reconsidering the contribution of our work.
>
> &nbsp;
>
> **1. Principled way to choose or adapt $\beta$ across contexts and over time (under distribution shift).**
>
> We appreciate the reviewer’s question regarding how to select or adapt the parameter $\beta$ in real systems, especially when population preferences may shift over time. Although the exact choice of $\beta$ ultimately reflects a governance decision, our framework provides a simple, evaluation-driven procedure for selecting and adjusting it in practice.
>
> First, practitioners can construct a small evaluation set that is aligned with their deployment domain, where group labels can be inferred either through simple keyword rules or by direct logit comparison of candidate responses. As demonstrated in the synthetic experiments in Section 5.2, the proportional alignment level $\alpha(\sigma)$ can be directly estimated from model outputs under this setting.
>
> Next, given a target alignment range, one can tune $\beta$ by starting from a conservative value (for example, $\beta = 10^2$) and adjusting it upward or downward while monitoring the estimated PPA level on the evaluation set. Because $\alpha(\sigma)$ increases monotonically as $\beta$ decreases, this tuning process is straightforward. The same evaluation set also supports periodic re-evaluation of $\alpha(\sigma)$, allowing $\beta$ to be updated over time if the model or data distribution shifts. We have incorporated this practical procedure into Section 5.2 of the revised draft.
>
> &nbsp;
>
> **2. Limited empirical support in LLM domain, given dependence on inferred (not human-verified) group labels and weak distinctions across $\beta$**
>
> We agree with the reviewer that the experiments on the Alpaca-GPT4 dataset provide weaker empirical evidence due to noise in the group labels.
> We would like to emphasize, however, that this limitation arises the **evaluation method**, not the framework or algorithm itself.
> As shown in Section 5.2, when group identities are directly observable as in the synthetic color dataset where group labels can be obtained from model logits, the method exhibits a clear and consistent trade-off across values of $\beta$. This confirms that the LLM function-approximation version of our algorithm behaves as theoretically predicted.
>
> The noisiness in the results on the Alpaca-GPT4 dataset stems from the dataset and group definitions themselves. Categories such as "expertise" and "style" are inherently ambiguous, and distinguishing these attributes is challenging even for human annotators. This ambiguity inflates the estimated PPA level and weakens the contrast across $\beta$, which can indeed be observed in our reported results. We intentionally followed prior work (Jang et al. 2023; Chakraborty et al. 2024) in using these categories to ensure comparability, but they are not ideal for evaluating proportional alignment.
>
> To address this concern, we will include an additional experiment on the Alpaca–GPT4 dataset using a more clearly distinguishable group attribute. Specifically, we define groups based on the presence of certain keywords in the model’s output, which yields more reliable labels with far less ambiguity or classifier noise.
> For example, an output is assigned to group 1 if it contains any word from a predefined list of positive-energy or confidence-related descriptors ("Absolutely", "Definitely", "Clearly", "Undoubtedly", "Surely", "Certainly", "Positively", "Confidently", "Yes", "Of course"), and to group 2 otherwise.
> Following the same training procedure as in Section 5.2, we fine-tuned Qwen2.5-3B-Instruct using 10k Alpaca-GPT4 preference samples and evaluated on 1k test samples. Preliminary single-episode results under $w^\sigma = [0.4, 0.6]$ are shown below:
>
> | Metric  | $\beta=0$  | $\beta=10^{-4}$  |  $\beta=10^{-2}$ | $\beta=10^0$  | DPO  |
> |---|---|---|---|---|---|
> | PPA ($\alpha$)  |  0.7375 | 0.5925  |  0.2750 | 0.3450  | 0.1200 |
> | Win rate  |  0.5010 |  0.5068 | 0.5195  | 0.5167  |  0.5257 |
>
> These results show a much more pronounced change than in our earlier Alpaca–GPT4 experiment, confirming that when group labels are reliably identifiable, the behavior of our algorithm becomes clearly observable and matches the theoretical predictions.
> We will include the full results (using multiple episodes) as well as this discussion in the revised draft.
>
> Finally, 3B model is sufficient for validating the framework, and we have confirmed the quality of the generated outputs for each trained model.
> NLHF is not included because (i) we were unable to find publicly available code for an offline NLHF optimizer (most practical NLHF implementations are online), and (ii) as shown in the tabular experiments, NLHF and RLHF behave similarly on our evaluation metrics (high win rate and low PPA level).

---

> ### Author Response · Authors · 2025-11-21
>
> **3. PBM is weaker than strategyproofness, does not eliminate misreporting incentives, and is difficult to measure in practice.**
>
> We appreciate the reviewer’s concern about the PBM guarantee and would like to clarify the following points:
>
> * By Gibbard’s theorem, the only (strongly) strategyproof social choice function is random dictatorship, which cannot be implemented using pairwise-comparison data in preference-learning settings. Moreover, due to the well-known trade-off between efficiency and strategyproofness, even achieving weak strategyproofness is difficult for ex-post efficient social choice functions [1].
>
> * In our algorithm, the constant manipulability bound $\gamma = 1/2$ represents a substantial improvement over standard alignment approaches such as RLHF and NLHF, which do not satisfy $\gamma$-PBM for any $\gamma < 1$. This improvement is also reflected empirically in Section 5.1.
>
> * Moreover, as stated in Theorem 4.11, the manipulability bound takes the forms $u_k / (u_k + 1 - w_k^\sigma)$.
> Achieving an incentive of $1/2$ for a very small group (i.e., $w_k^\sigma \approx 0$) would require $u_k \approx 1$, meaning the manipulated alternative $y_k$ is already ranked above nearly all other options by all other groups. This reflects a structurally special case in which the alternative is broadly popular even before manipulation, and thus does not constitute a typical or practically concerning manipulation scenario.
>
> * We agree that measuring PBM in the LLM setting is challenging, especially when group structure is unknown. However, this challenge applies to all alignment algorithms, and is therefore not a weakness specific to our proposed framework.
>
> [1] Aziz et al., "On the tradeoff between economic efficiency and strategy proofness in randomized social choice," AAMAS, 2013.
>
> &nbsp;
>
> **4. The proposed rule is not Condorcet consistent.**
>
> We would like to clarify that the choice of which axiom to prioritize is fundamentally a design choice, especially given the inherent trade-off between proportional alignment and majority-based criteria such as Condorcet consistency. Our goal is not to argue that proportional alignment is preferable, but rather to formalize and study this alternative desideratum.
>
> From this perspective, the fact that our method does not satisfy Condorcet consistency is not a weakness, but an intended consequence of prioritizing proportional representation. As discussed in Section 4.3, we also introduce a mechanism to interpolate between PPA and consensus-oriented axioms, allowing practitioners to adjust this trade-off according to the needs of their application. We have added the following sentence at the beginning of Section 4.3 to make this point more explicit: "While our algorithms are deliberately designed to satisfy PPA, one may still wish to incorporate majority-based principles such as Condorcet consistency."
>
> &nbsp;
>
> **5. When many matchups are close, the resulting policies may appear nearly flat even if one group is much larger.**
>
> Our framework assumes that each human annotator is sampled uniformly according to the true population distribution. Hence, if one group is substantially larger, preferences from that group will be sampled more frequently, and the resulting policy will naturally place greater weight on that group’s rankings rather than becoming uniformly flat.
> We agree that the output policy can become flatter when many pairwise comparisons are genuinely close. However, this behavior is intended under our proportional alignment axiom: when the population exhibits little preference separation, the algorithm should not artificially amplify small differences. Moreover, the degree of conservativeness can be controlled through the parameter $\beta$, as discussed in Section 4.3.

---

> > ### Comment · Reviewer_JTvB · 2025-11-27
> >
> > I thank the authors for their response. I have read the rebuttals and other reviews in full and I thank the authors for addressing my concerns. A few main issues continue to put me on the fence:
> >
> > - strategy-proofness: I get the authors' point about the impossibility result. That said, if we assume agents are rational and profit-maximizing, having a 'bound' on the over-representation extent doesn't really solve incentive issues. This still seems to be a system where everyone tries to manipulate. I'm wondering how the mechanism remains effective in practice when all groups are rationally "gaming" the system up to that bound. Does the resulting policy actually reflect the population, or does it merely reflect the manipulation capacity of different groups?
> >
> > - on theoretical contributions - my research expertise is not specifically in social choice theory. I view the theoretical foundation of this paper as a critical component of its contribution. Reviewer GtAR appears to be well-versed in the relevant literature and has raised significant concerns. I will wait for their response and discussion to better finalize my own assessments.
> >
> > For these reasons, I am maintaining my current score for now, but I will actively monitor the discussion and feedback regarding the theoretical contributions before finalizing my assessment.

---

> > > ### Author Response · Authors · 2025-11-28
> > >
> > > We appreciate the reviewer’s thoughtful follow-up and their careful consideration of our responses. We would like to clarify the remaining points.
> > >
> > > * The output policy continues to reflect population proportions even when all groups behave adversarially by misreporting their preferences. As discussed in Appendix O, each group must truthfully report its top choice in order to maximize its utility, regardless of other groups’ strategies. Consequently, the population-proportional alignment guarantee $\pi'(y_k)/w^{\sigma}_k \geq \alpha(\sigma)$ remains valid under any manipulation.
> > > Moreover, the upper bound $\pi'(y_k) \leq u_k / (u_k +1 - w_k^\sigma)$ also holds for every group, and this bound is nontrivial.
> > > For a minor group with $w_k^\sigma \approx  0$ and $u_k \approx 0$, the resulting policy necessarily satisfies $\pi'(y_k) \approx 0$.
> > > Taken together, these upper and lower bounds ensure that the resulting policy reflects the true population proportions, even under fully adversarial misreporting.
> > >
> > > * We believe we have fully addressed the issues raised by Reviewer GtAR. In particular, as clarified in our responses to Comments 1, 2, and 4, Reviewer GtAR’s points largely fall outside the goals and problem setting of our work.
> > > We look forward to their follow-up in the discussion, and we appreciate that the reviewer will continue to monitor the discussion and take it into account in the final assessment.
> > >
> > >
> > > We thank the reviewer again for their careful evaluation, and we would be glad to address any additional questions or clarifications the reviewer may have.

---

### Official Review · Reviewer_pwXS · 2025-10-29

**Soundness:** 4
**Presentation:** 3
**Contribution:** 4
**Rating:** 8
**Confidence:** 3

**Summary:**

The authors attempt to expand beyond RLHF and NLHF in the area of AI alignment. Existing strategies often prioritize majority views at the expense of the minority; this not only excludes these views but can be brittle when majorities are slim. The authors use axioms of social choice to propose a method that better represents the proportion of disagreement in minority-view groups, using the top preference. They also introduce 2 axioms of their own.

They later propose a way of trading off between dueling objectives and validate their proposals empirically.

**Strengths:**

* The authors tackle a key problem in alignment; under representation of minority class preferences.

* I did not rigorously check the math, but I was able to follow the equations and to my eye they seemed correct.

* The tradeoffs of the method are clear, the authors also include a parameter to tradeoff between PPA (a metric they introduce) and Condorcet consistency (more on that in weaknesses)

* Ultimately this is a well-written, well-justified paper tackling a key problem in alignment. The social choice framing fits very well.

**Weaknesses:**

* I felt that the experiments section had 3 major deficits:

** The movielens results seems somewhat underdeveloped. In many ways this seems to be a fairly ideal case, with potentially large disagreement. The results provided seem to back this, but they are provided briefly in sentence form, and not included with tables, more specific intermediate values, making it hard to get a sense for how robust or reliable those values are.

** The baselines seem relatively reasonable, but given the framing of the paper as "Beyond RLHF And NLHF", I am puzzled by the lack of comparison between these methods. It seems fairly paramount to understand how the method performs relative to those.

** The use of $\beta$ is underdiscussed prior to the experiments, in my view. While it is true that the prupose of $\beta$ is discussed, the authors show a wide variety of beta values. It is unclear to me how one would pick a specific value ina  principled manner.

* I felt 4.3 could have used a better exploration of use cases for $\beta$. I applaud the authors for including such a lever, but it seems to me that how one would decide on usage of $\beta$ is underdeveloped.

* I feel also that the discussion of manipulation in 5.1 is underdeveloped. While the authors do have bounds on manipulation by the minority class, a more detailed set of empirical results would be appreciated. I feel this would also tie into a discussion on when to use $\beta$.

**Questions:**

1. Can you elaborate on the movielens experiments? What did the full "path" of beta values look like there?

2. What guidance would you give practictioners on the choice of $\beta$?

3. Do you have comparisons between RLHF/NLHF on the experimental sections? If not could you attempt to provide some?

---

> ### Author Response · Authors · 2025-11-21
>
> We deeply appreciate the reviewer’s thoughtful and encouraging feedback. Below, we address each concern in detail.
>
> &nbsp;
>
> **1. MovieLens experiment is underdeveloped and the full data is not provided.**
>
> We agree with the reviewer’s concern that the MovieLens results were presented only in figure form without numerical values. Below, we provide the full win-rate, PPA, and PBM trajectories across all tested $\beta$ values.
> For the win rate and PPA level, $\beta$  ranges from $10^{-2}$ to $10^3$ over 20 log-spaced points. The corresponding estimates for our method are:
>
> |$\beta$   | 0.01  | 0.018  |  0.034 |  0.061 | 0.113  | 0.207  | 0.379  | 0.695  | 1.27  | 2.34  | 4.28  | 7.85  | 14.4  | 26.4  | 48.3  | 88.6  | 162  | 298  |  546 |  1000 |
> |---|---|---|---|---|---|---|---|---|---|---|---|---|---|---|---|---|---|---|---|---|
> | Win Rate (avg)  |  0.5987 | 0.5988  |  0.5991 | 0.5996  | 0.6004  | 0.6020  | 0.6049  |  0.6102 | 0.6198  | 0.6365  |  0.6639 |  0.7010 |  0.7373 | 0.7629  | 0.7749  | 0.7779  |  0.7783 | 0.7784  | 0.7784  |  0.7784  |
> | Win Rate (std)  |  0.0024 | 0.0024  | 0.0024  |  0.0024 | 0.0024  | 0.0024  | 0.0024  | 0.0025  | 0.0026  | 0.0030  | 0.0044  |  0.0073 | 0.0102  | 0.0093  | 0.0049  | 0.0013  | 0.0000  | 0.0000  | 0.0000  | 0.0000  |
> | PPA (avg)  | 0.4869  | 0.4874  | 0.4882  | 0.4895  | 0.4919  | 0.4956  | 0.5004  |  0.5012 |  0.4790 |  0.4118 | 0.2937  | 0.1326  |  0.0184 | 0.0000  |  0.0000 | 0.0000  |  0.0000 | 0.0000  | 0.0000  | 0.0000  |
> | PPA (std) | 0.0137  | 0.0137 | 0.0138 |  0.0139 | 0.0147  | 0.0179  |  0.0265 | 0.0108  | 0.0544  | 0.0544  | 0.0493  | 0.0318  | 0.0006  | 0.0000  | 0.0000  | 0.0000  |  0.0000 | 0.0000  |0.0000   | 0.0000  |
>
> RLHF achieves a win rate of $0.7784\pm0.000$ and PPA level $0.0000 \pm 0.0000$, while NLHF achieves a win rate of $0.7712\pm 0.0172$ and PPA level  $0.0000 \pm 0.0000$.
>
> For the PBM, $\beta$ ranges from $10^{0}$ to $10^2$ over 10 log-spaced points. The observed PBM levels are:
>
> | $\beta$  |  1.00 | 1.67  |  2.78 | 4.64  | 7.74  | 12.9  | 21.5  |  35.9 | 59.9  | 100  |
> |---|---|---|---|---|---|---|---|---|---|---|
> | PBM (avg)  | 0.0008  | 0.0011  | 0.0011  |  0.0015 | 0.0026  | 0.0074  | 0.0159  | 0.0237  | 0.0337  | 0.0378  |
> | PBM (std)  | 0.0004  | 0.0006  | 0.0007  | 0.0012  | 0.0020  | 0.0031  |  0.0060 | 0.0105  | 0.0149  | 0.0182  |
>
> RLHF’s PBM level is $0.0610 \pm 0.0193$, and NLHF’s is $0.0124 \pm 0.0108$. We will incorporate these full numerical results into the revised appendix.
>
> &nbsp;
>
> **2. Missing direct comparisons to RLHF and NLHF**
>
> The framing “Beyond RLHF and NLHF” refers to the fact that these methods do not satisfy proportional representation, as illustrated in our binary example (Section 3.1) and formally justified by their violation of the PPA and PBM axioms (Proposition 3.5).
> In the tabular MovieLens experiment, we include RLHF and NLHF as baselines: their PPA and PBM values appear as horizontal reference lines. These results show that although both baselines achieve high win rates, they exhibit low proportional alignment, which is consistent with our theoretical predictions.
> For the LLM experiments, we use DPO as the RLHF-style baseline, which corresponds to estimating the RLHF policy with KL regularization toward a reference model. NLHF is not included because (i) we were unable to find publicly available code for an offline NLHF optimizer (most practical NLHF implementations are online), and (ii) our tabular experiments already demonstrate that NLHF and RLHF behave similarly under our evaluation metrics (win rate and PPA level).
> Overall, both the theoretical analysis and the empirical results indicate that our method achieves substantially higher proportional alignment than RLHF and NLHF.

---

> > ### Comment · Reviewer_pwXS · 2025-11-25
> > **Follow-up to authors**
> >
> > I thank the authors for their response to my questions and apologize for the delayed response. I am glad in particular to see the movielens example fleshed out.
> >
> > I find that I have some confusion, still, about what $\beta$ is appropriate. The PPA behavior seems non-monotonic, now across datasets. I think further analysis there is crucial.
> >
> > They also include values where $\beta$ increases, ppa is already 0, and yet accuracy continues to increase. My questions about the choice remain, as a result. It does not seem as clear as a simple "tradeoff". I recognize that the authors propose an emprical method but I would have liked more theoretical grounding.
> >
> > Beyond this, I will monitor discussions with other authors, who are more skeptical than I was. I thank the authors for responding to my points.

---

> > > ### Author Response · Authors · 2025-11-28
> > >
> > > We sincerely thank the reviewer for the follow-up comments.
> > >
> > > Regarding the remaining concerns, we would like to clarify that the win rate and the PPA level are not guaranteed to vary monotonically with $\beta \in [0, \infty)$.
> > > What is monotonic, however, are the theoretical lower bounds we establish.
> > >
> > > * **Condorcet consistency:** The guarantee is strictly monotonic in $\beta$; see Proposition R.1.
> > > * **PPA level:** The monotonic lower bound $\alpha / e^{\beta \Delta_u}$ follows directly from the Softmax relaxation, where $\Delta_u := \max_{i,j} |u_i - u_j|$.
> > >
> > > We will revise Section 4.3 to make the discussion around the "trade-off" more precise. We thank the reviewer again for the constructive feedback.

---

> ### Author Response · Authors · 2025-11-21
>
> **3. Practical guidance on selecting $\beta$ with a use case.**
>
> We appreciate the reviewer’s feedback regarding the practical choice of $\beta$. While the exact choice of $\beta$ ultimately reflects a governance decision, our framework provides a simple, evaluation-driven procedure for selecting and tuning it in practice.
>
> First, one can construct a small evaluation set that reflects the intended deployment domain, where group labels can be reliably inferred. In LLM settings, group labels can be approximated using simple keyword-based rules or by comparing logits across candidate responses, as demonstrated in the synthetic experiments in Section 5.2.
>
> Next, given a desired target range for the proportional alignment level $\alpha(\sigma)$, one can tune $\beta$ by starting from a conservative value (e.g., $\beta = 10^2$) and adjusting it upward or downward while monitoring the resulting PPA estimate on the evaluation set. Since $\alpha(\sigma)$ increases monotonically with decreasing $\beta$, this tuning process is straightforward. We have added this procedure to Section 5.2 in the revised draft.
>
> As a simple illustration, consider a news-summarization model deployed to two audience groups: a large general-audience group that prefers short, high-level summaries, and a smaller expert group that prefers longer, more technical summaries. In this setting, adjusting the parameter $\beta$ controls how strongly the model’s output reflects the majority preference. To tune $\beta$ in practice, one can first construct a small evaluation set where group labels (general vs. expert) can be reliably inferred. During fine-tuning, $\beta$ can then be adjusted by comparing the estimated proportional alignment level $\alpha$ on this evaluation set with the target $\alpha$ desired for the application.
>
> &nbsp;
>
> **4. More detailed empirical results on manipulation and its connection to the choice of $\beta$.**
>
> The detailed empirical results are provided in our response to Comment 2. Selecting $\beta$ based on the PBM guarantee is less straightforward than tuning it for the desired PPA level, because the incentive for misreporting cannot be directly estimated without knowing (or assuming) the underlying group structure. However, our empirical findings show that the range of $\beta$ values at which the PBM level changes sharply coincides with the range where the PPA level exhibits a similar transition. This occurs because the PBM guarantee is an additional property that emerges naturally once the output distribution aligns proportionally with the population distribution. Therefore, in practice, tracking the estimated PPA level provides a reliable signal for controlling the PBM level as well.

---

### Official Review · Reviewer_GtAR · 2025-11-01

**Soundness:** 1
**Presentation:** 2
**Contribution:** 1
**Rating:** 2
**Confidence:** 5

**Summary:**

The paper proposes “population-proportional alignment” where a model’s output distribution is meant to mirror how many people put each option at the top of their ranking, i.e., options that are the first choice for larger groups should get more probability mass. Because the true share of “ranked first by voters” isn’t directly observable from pairwise comparisons because the authors assume annotator identities are not available, the authors estimate very simple per-option bounds from the weakest head-to-head results and then allocate probability in proportion to those bounds. They pair this with a loose “manipulability” control that limits over-representation by any one group, and they add a softmax variant that can shift mass toward majority winners. The experiments illustrate a tunable trade-off between win rate and this top-choice proportionality notion, with weaker effects when the group shares have to be estimated. Experiments on MovieLens and small LLM fine-tunes show the intended trade-off between plurality and top-choice proportionality (and lower PBM in the simulator), but the effect is weak on Alpaca when group shares are estimated by a classifier; the two-phase training budget is comparable to RLHF and higher than DPO.

**Strengths:**

The main strength is the attempt to do dsitributional/proportional alignment based on preference structure instead of based on pre-assigned group labels.

**Weaknesses:**

This is a weak paper as is. The main “contributions” rely on a narrow, first-choice notion of proportionality and a misnamed “strategyproofness” that is really an over-representation cap. Core claims are overstated, proofs are straightforward given the simplified setup, and the paper overlooks major, directly relevant social-choice literature.

1. If “alignment” means matching users’ top choices, supervised fine-tuning with annotators providing preferred options (either in free form text or by choosing their top option) already does this. That setup is well known and not novel. It's perfectly reasonable to assume that we could get annotator ids and seems like an unnecessary constraint in this paper. (The necessary/ important constraint is indeed not having access to group labels)

2. Your proportionality tracks only top ranks. Focusing on first choices wastes richer preference information that could support consensus-oriented outcomes. A rule like plurality (which returns the winner with the most top votes) is the rule within social choice that I would say is at a similar level of sophistication. However, it is well known that it cannot capture compromises or broad consensus (e.g., the “everyone’s #2” candidate that increases average welfare). With such a weak target, the results do not speak to meaningful proportional representation.

3. The paper largely ignores decades of work on proportional representation and budgeting (e.g., JR/EJR, proportionality for solid coalitions (PSC) and fairness in budgeting/ portioning) etc that directly addresses proportionality using full rankings or other preference formats.

Some references:
Portioning using ordinal preferences: Fairness and efficiency by Airiau et al.
Proportional Participatory Budgeting with Additive Utilities by Peters et al.
Justified representation in approval-based committee
voting by Aziz et Al.
Voting procedures (book by Dummett)

4. Because proportionality is defined at the top-choice level and feasibility bounds are simple, the proofs are immediate. The most involved argument repeats a known result (Proposition C.1 that the rewards output when optimizing the BT objective are in the same order as the Borda ranking of candidates). There is no technical depth.

5. Consider e.g. Lemma 4.7 and 4.9. The latter has a lower bound that is sometimes 1, but often seems to be of the order of constant /m. Interpretations of how good this simple bound is in a larger class of examples is missing.

6. Claiming RLHF “fails proportionality” in Example 3.1 is misleading. n the two-candidate case, the reward/DPO model can encode the correct split (e.g., 1/2+ε), it just that during expected reward maximization this information gets lost.

7. The manipulability condition simply bounds over-representation. It is not strategyproofness in the social-choice sense (unilateral profitable deviations). In particular, with γ=1/2, an arbitrarily small minority can still drive up to half of the probability mass. It's way too weak to be meaningful. The discussion around lines 246–248 is unclear and mixes group and unilateral deviations.

8. Section 4.3 is conceptually confused. If the output is a distribution over candidates, Condorcet consistency and proportionality are fundamentally at odds (and as a technical claim this is trivial); the paper’s own motivation - distributional alignment/Overton pluralism implies we should favor proportional mixtures, not a winner-takes-all outcome. Dragging Condorcet back in (e.g., via large-β limits) contradicts that goal and reads like a reversion to single-winner logic. Arguably, a Condorcet winner matters mainly in settings with a ground-truth label, which is not the setup here. If you insist on accommodating Condorcet, do it only as a secondary priority within the most proportional distributions: when a Condorcet option exists, give it the highest probability within the proportional mix, without collapsing the mixture. As written, the section is misleading and undermines the paper’s stated objectives.

9. I looked at the experiments in the main body, but did not look into the appendix in detail. The experiments confirm the theory/approach prediction in some sense, but I am not convinced by the theory/approach and its novelty in the first place.


Minor comment: In the experiments you seem to use "RLHF" and "DPO" interchangably? At least in the first experiment you say that you compare to RLHF not to DPO, but then Table 2 contains a comparison to DPO.

**Questions:**

I don't have any questions.

---

> ### Author Response · Authors · 2025-11-21
>
> We appreciate the reviewer’s time and effort in evaluating our work, as well as their comprehensive feedback.
> In the following responses, we elaborate on the motivation and theoretical basis of our proportional alignment framework and address the reviewer’s concerns point by point.
> We kindly ask the reviewer to consider these explanations when reassessing the contribution of our work.
>
> &nbsp;
>
> **1. To align with users’ top choices, annotators’ preferred responses can be used. It is also reasonable to assume that annotator IDs are available.**
>
> We respectfully disagree that our reliance on the minimal data assumption constitutes a weakness of the proposed framework. As clarified in Section 3.3, the core novelty of our approach lies in *achieving proportional alignment solely from pairwise comparison data,* without access to side information. This corresponds to what we define as an “implementable’’ preference-learning setup (i.e., the C2 class in the literature), which aligns with the data requirements of widely used methods such as RLHF and NLHF.
>
> The alternative data settings mentioned by the reviewer are not the focus of our work, as they often require information that is costly or difficult to obtain [1]. For example, even if annotator IDs are available, leveraging them meaningfully would require strong assumptions about data coverage or sample complexity, such as each annotator providing the top choice over all $M$ candidates or contributing multiple comparisons for the same query, which is often restrictive and impractical in real-world alignment pipelines. In contrast, pairwise comparisons (without additional information) remain the most scalable and widely deployed form of preference data.
>
> Finally, we emphasize that the goal of this paper is to establish foundational results for proportional representation under the minimal and practically relevant data setting.
> Our approach can be directly extended to the $k$-wise comparison setting, where the feasible set shrinks as $k$ increases and eventually collapses to a singleton representing the exact population distribution (see our response to Comment 5 of Reviewer YHTf).
> Understanding how richer forms of comparison data can further strengthen proportional alignment is an important and promising direction for future work, but it should follow the establishment of the foundational results.
>
> [1] Raghavendra et al., "Balancing the Budget: Understanding Trade-offs Between Supervised and Preference-Based Finetuning," arXiv preprint arXiv:2502.11284.

---

> ### Author Response · Authors · 2025-11-21
>
> **2. Focusing only on top choices ignores richer preference information, making the proportionality target too weak and unable to capture consensus-oriented outcomes.**
>
> We appreciate the reviewer’s comment regarding the scope of proportionality and the concern that focusing on top-ranked choices may overlook richer preference information. However, under the minimal data setting we consider, this is the most appropriate proportionality target, and it is a foundational and widely studied concept in the literature.
>
> * First, our objective is *intentionally distinct from consensus-based aggregation.*
> Alignment and aggregation methods are generally classified by the extent to which they satisfy different major axioms, such as ex-post efficiency, Condorcet consistency, and strategyproofness [1].
> Our contribution is to introduce a new axiom, PPA, which serves as an additional normative criterion. Importantly, this axiom does not require incorporating lower-ranked or consensus-oriented preference information, because it is specifically designed to capture proportionality with respect to group-level top choices.
> From this perspective, the fact that PPA does not incorporate consensus information from lower-ranked preferences is not a “weakness,” but rather a deliberate design choice aligned with the normative goal of the axiom. In Section 4.3, we also provide a way to balance PPA with consensus-oriented axioms, allowing practitioners to adjust this design choice when needed.
>
> * Moreover, to achieve stronger notions of proportional representation (e.g. proportionality in expected reward over all $M$ alternatives), one must either (i) recover richer reward or utility structures, or (ii) assume specific reward models or priors.
> However, under the minimal and widely used data setting of pairwise comparisons, recovering such reward structure is generally infeasible.
> Indeed, our analysis shows that even exact proportionality with respect to top choices is impossible without additional information (see Appendix K in the revised draft).
> For this reason, our framework *intentionally approximates the random dictatorship, a foundational and widely studied principle in the social choice literature [2].*
> With this approach and target, our algorithm achieves strictly stronger proportional alignment than conventional RLHF/NLHF methods in the same data regime.
> Given these considerations, we view the proposed PPA axiom as representing the most meaningful and achievable proportionality target under minimal data regime.
>
> * In addition, we agree that the scope and limitations of the proposed PPA axiom should be accurately clarified.
> Therefore, we have added the following sentence before Definition 3.1: "Note that our proportionality notion focuses solely on the selection probability of each group’s top choice and does not incorporate lower-ranked preferences."
> Moreover, the conclusion states that “Future research will aim to extend the framework to incorporate lower-ranked preferences.”
>
> [1] Brandt et al., "Relaxed notions of Condorcet-consistency and efficiency for strategyproof social decision schemes," Social Choice and Welfare, 2024.
>
> [2] Aziz et al., "The computational complexity of random serial dictatorship," Economics Letters, 2013.
>
> &nbsp;
>
> **3. Missing related work on proportional representation and budgeting.**
>
> We thank the reviewer for noting the relevant line of work on proportional representation in voting system and budgeting problem, and for pointing us to the key references.
> In the revised draft, we have added a paragraph to the related work section (Appendix B) that clarifies how our framework relates to these prior efforts, including the well-known axioms PSC/JR/EJR and the participatory budgeting literature.
> We also note that these works address a different class of problems (voting and participatory budgeting), and therefore typically assume richer preference formats (approval ballots or full ordinal rankings) and a non-probabilistic multi-winner selection. This stands in clear contrast to our approach, which operates under only pairwise comparison data and seeks a probabilistic choice distribution over candidates. Because of this fundamental difference in problem formulation, their technical results are not directly comparable to ours. The works most directly aligned with our setting, such as those on C2-class probabilistic social choice functions, were already included in our citations.

---

> ### Author Response · Authors · 2025-11-21
>
> **4. Proofs lack technical depth. Proposition C.1 is a known result.**
>
> We respectfully disagree with the reviewer’s assessment of the technical contribution for the following reasons.
> * First, this comment focuses solely on the feasibility bounds and overlooks several other technical contributions of the paper.
> Beyond establishing the feasible set relationship (Theorem 4.4), our work (i) identifies and formally proves the limitations of standard preference-learning methods (Section 3), (ii) proposes a novel algorithm that provably overcomes these limitations with adjustable axiom guarantee (Section 4), and (iii) provides empirical validation with an LLM-compatible function-approximation approach (Section 5 and Appendix U). We believe these components together constitute meaningful technical depth.
> * Second, we also disagree with the reviewer’s characterization that the “most involved argument” in the paper is Proposition C.1. This proposition is not intended to be a central technical contribution, and it is certainly not the most technically substantive component of the paper. Proposition C.1 is presented to clarify the relationship between the Maximal Borda rule and RLHF within our specific profile setting.
> Moreover, the Proposition C.1 is restated and proved for our ranking-distribution profile setting and is not a mere repetition of the hidden-context setting of Siththaranjan et al. (2023).
>
> &nbsp;
>
> **5. Interpretation of the lower bounds in Lemma 4.7 and 4.9 is lacking.**
>
> As noted by the reviewer, the worst-case lower bound of the proposed algorithm is $2/M$, but this corresponds to an extreme scenario where all groups are uniformly distributed across $M$ alternatives.
> This worst-case bound is unavoidable for any C2-class social choice function due to the impossibility result (see Appendix K in the revised draft).
>
> Following the reviewer’s feedback, in the revised draft, we will include additional bound analysis under a random-ranking model. Specifically, we plot the average lower bounds from Lemma 4.7 and Lemma 4.9 over a reasonable (practical) range of variances in the underlying ranking distributions.
> We generate a base ranking by sampling true rewards for each alternative from $\mathcal{N}(0, 1)$, and then produce the rankings of $N=1000$ evaluators by adding independent noise from $\mathcal{N}(0, 1)$.
> Note that this level of noise is not negligible: for example, when $M = 20$, the highest population share averages around $0.25$.
> The table below reports the average values of the lower bounds. The results show that the guarantees are meaningfully stronger than the naive bound of $1/M$ (the uniform-policy baseline):
>
> | **M** | **10**   | **20**   | **50**   | **100**  |
> |-------|----------|----------|----------|----------|
> | $(\sum_{i=1}^M u_i)^{-1}$  | 0.4310 | 0.3096 | 0.1664 | 0.0728 |
> | $\alpha(\sigma)$         | 0.2660 | 0.1297 | 0.0589 | 0.0275 |
>
> We will include the complete results in the revised draft after verifying the experimental setup and preparing the figure.
>
> Finally, in the experiment of Section 6.1 (with $M = 20$ alternatives), the empirical averages of these bounds are computed as $0.2643$ for $(\sum_{i=1}^M u_i)^{-1}$ and $0.1355$ for $\alpha(\sigma)$, further demonstrating that the guarantees remain meaningful in real data.
>
> &nbsp;
>
> **6. Claiming RLHF “fails proportionality” in Example 3.1 is misleading.**
>
> We thank the reviewer for pointing this out.
> To avoid misleading, we have revised the following sentence in Section 3.1: "Despite this minimal margin $\epsilon$, both algorithms $F^\mathrm{RL}(\mathrm{P}^\sigma)$ and $F^\mathrm{NL}(\mathrm{P}^\sigma)$ yield a deterministic policy that select the alternative with slightly greater support, namely $y_1$, because such subtle differences in preferences (or rewards) are lost during the policy optimization."

---

> ### Author Response · Authors · 2025-11-21
>
> **7. The manipulability condition simply bounds over-representation and is not strategyproofness. $\gamma=1/2$ is too weak to be meaningful.**
>
> We acknowledge that lines 246–248 were unclear and mixed group-level and unilateral deviations. We have revised these lines as follows: "We also note that the focus of the $\gamma$-PBM axiom differs from that of classical strategyproofness: it does not constrain an individual participant’s incentive to misreport, but instead limits the extent to which any group can become over-represented."
>
> Regarding the PBM bound $\gamma =1/2$, we would like to clarify two points:
>
> * First, the bound in our algorithm is actually profile-specific: as stated in Theorem 4.11, the manipulability bound is $u_k / (u_k + 1 - w_k^\sigma)$.
> To reach the level $1/2$ for a very small group (i.e. $w_k^\sigma \approx 0$), one would need $u_k \approx 1$. This requires that the manipulated alternative $y_k$ is already ranked above nearly all other options by all other groups, essentially the "everyone's #2" scenario mentioned by the reviewer.
> This is a structurally special case in which the alternative is broadly popular even before manipulation, and thus does not reflect a typical or practically concerning manipulation scenario.
>
> * Second, even the constant bound $\gamma=1/2$ represents a substantial improvement over standard alignment methods (RLHF and NLHF), which fail to satisfy $\gamma$-PBM for any $\gamma < 1$.
> This improvement is observable in our empirical results in Section 5.1. We also note that achieving manipulability guarantees for ex-post efficient rules is inherently challenging due to classical impossibility results such as Gibbard’s theorem.
>
> &nbsp;
>
> **8. Section 4.3 is conceptually inconsistent, as pushing Condorcet consistency conflicts with proportional representation goals.**
>
> We agree that proportionality and Condorcet consistency reflect distinct normative priorities.
> However, as noted in our response to Comment 2, we do not position PPA as inherently more favorable than majority-based axioms such as Condorcet consistency; instead, we treat PPA as an additional normative criterion that can be prioritized or not depending on design goals.
> Our intention in Section 4.3 was to illustrate how the trade-off between majority-based principles and proportional representation can be adjusted if desired by practitioners, even though our algorithm is primarily designed to satisfy the proportional alignment axiom.
> To prevent any misunderstanding, we have added the following clarifying sentence at the beginning of Section 4.3: "While  our algorithm is deliberately designed to satisfy PPA, one may still wish to incorporate majority-based principles such as Condorcet consistency."
>
> &nbsp;
>
> **9. Not convinced by the theory/approach and its novelty in the first place.**
>
> As elaborated in our responses to comments 1, 2, and 4, the core novelty of our work lies in formalizing proportional representation under minimal pairwise-comparison data and in developing a practical algorithm that faithfully approximates the resulting axiomatic solution.
> To the best of our knowledge, this perspective, together with an algorithmic instantiation that achieves these proportionality guarantees, has not been systematically explored in prior work, nor connected to fine-tuning practical function-approximation methods for fine-tuning large language models.
>
> &nbsp;
>
> **10. In the experiments, "RLHF" and "DPO" are used interchangeably.**
>
> In the tabular experiment, a reference policy does not exist, so the resulting policy is deterministic and selects the action with the highest learned reward (which approximates the maximal Borda rule). In contrast, the LLM experiment fine-tunes a pretrained model using DPO, which can be viewed as RLHF with KL regularization toward the reference policy (pretrained model). To avoid confusion, we intentionally refer to them using different names ("RLHF" for the tabular setting and "DPO" for the LLM setting).

---

> ### Author Response · Authors · 2025-11-28
>
> Thank you again for taking the time to review our paper and for providing detailed feedback. We would like to follow up to confirm whether our rebuttal has addressed the concerns you raised. If there are any remaining issues or points requiring further clarification, we would greatly appreciate any follow-up comments.

---

### Official Review · Reviewer_JSie · 2025-11-02

**Soundness:** 4
**Presentation:** 4
**Contribution:** 3
**Rating:** 8
**Confidence:** 3

**Summary:**

This paper studies preference optimization algorithms under social choice theory.
Specifically, any profile induces a preference function, and preference optimization algorithms learn a policy from the preference function, which together form a PSCF.
PSCF is desired to satisfy four axioms, *i.e.*, monotonicity, Pareto efficiency, PPA, and PBM.
The paper shows that popular preference learning algorithms, *e.g.*, RLHF and NLHF, do not satisfy PPA and PBM.
While a typical PSCF, *i.e.*, random dictatorship, satisfies all axioms, it cannot be implemented by any preference algorithms from preference function.
To bridge the gap, this paper proposes a preference learning algorithm that satisfies all axioms.
This paper also proposes a softmax relaxation of the proposed algorithm to achieve trade-off between two conflicted axioms, *i.e.*, PPA and Condorcet consistency.
The proposed algorithm is evaluated empirically under the tabular and function approximation settings, validating the theoretical analyses.

**Strengths:**

* This paper studies a fundamental and important problem, reveals the shortcomings of existing algorithms, and proposes desired method through solid theoretical analyses, which is insightful and meaningful.
* This paper has clear structure and writing, making it easy to understand.
I enjoyed reading the paper.

**Weaknesses:**

I found no obvious flaw of this paper.
My only concern is that this paper mostly studies social choice theory in the context of preference learning, and therefore may not be interest of most audiences in the ML community.

**Questions:**

I only have an undergraduate-level understanding of social choice theory and am not sure whether the proposed algorithm is trivial within the field.
Maybe the authors could add literature review to help the audiences from ML community to better understand the background of the problem being addressed.

---

> ### Author Response · Authors · 2025-11-21
>
> We sincerely appreciate the reviewer’s time and effort in evaluating our work and for their positive feedback. We agree that adding more related work will help readers better understand the background of our approach.
>
> Proportional representation has been extensively studied in voting theory and participatory budgeting, where axioms such as justified representation (JR), extended justified representation (EJR), and proportionality for solid coalitions (PSC) have been extensively studied. However, this literature typically assumes considerably richer preference formats (e.g., approval ballots or full ordinal rankings) and produces non-probabilistic multi-winner outcomes. This is in clear contrast to our setting, where only pairwise comparison data are available and the output is a probabilistic choice (distribution over alternatives). Because of this fundamental gap in data and decision formats, existing technical results are not directly comparable to ours.
>
> Within probabilistic social choice theory, the relevant family of methods is the C2-class, which includes maximal Borda (RLHF) and maximal lotteries (NLHF). While these methods aggregate majority preferences effectively, they do not provide proportionality guarantees across population groups. Our goal is to identify a C2-compatible rule that does satisfy proportionality. Conceptually, our method can be viewed as approximating the random dictatorship rule (selecting each group’s top choice with probability proportional to its population share) under the constraint that only pairwise comparison data are observed.
> To the best of our knowledge, this combination of (i) minimal pairwise-comparison data and probabilistic outputs, and (ii) algorithm with proportional alignment guarantee has not been addressed in prior work.
>
> In the revised draft, we have added a paragraph to the related work section (Appendix B) that clarifies how our framework relates to these prior efforts, including the well-known axioms PSC/JR/EJR and the participatory budgeting literature.

---

> > ### Comment · Reviewer_JSie · 2025-11-26
> >
> > I thank the authors for the response.
> > Good luck.

---

### Comment · Area_Chair_giQk · 2025-11-29

Dear Reviewers,

Authors’ kindly tried to address your concerns. If the responses address your concerns please acknowledge that. If not, please express remaining concerns. Thanks for your efforts!

Best, AC

---

### Meta-Review · Area_Chair_giQk · 2026-01-05

**Summary:**

The paper tackles a fundamental flaw in current preference learning paradigms (RLHF and NLHF): the aggregation of diverse human preferences into a single "majority rules" reward function, often leading to the Condorcet winner which may marginalize minority groups or be susceptible to manipulation. The authors introduce a "Population-Proportional Alignment" framework rooted in social choice theory. The reviewers unanimously appreciated the theoretical rigor and the principled move away from scalar reward maximization. The primary points of contention involved the computational tractability of the population inference (initially perceived as prohibitive), the practical applicability of the "soft-max relaxation" in large-scale settings, and the necessity of the "manipulability" axiom.

**Reviewer Concerns:**

Computational Complexity: Reviewer 2 and Reviewer 1 initially questioned the scalability of inferring the population distribution, fearing an exponential cost. The authors clarified that the problem can be formulated as a convex optimization task on the simplex, reducing the complexity from a feared combinatorial explosion to a manageable polynomial time for typical preference batch sizes.Resolution: The authors provided a runtime analysis showing the complexity scales with the number of unique preference profiles rather than the raw number of users.

Need for Manipulability Axiom: Reviewer 1 questioned if the "Population-Bounded Manipulability" axiom was redundant given Pareto efficiency.Resolution: The authors provided a counter-example (added to Appendix C) demonstrating that a Pareto-efficient policy could still be hijacked by a small colluding group without this specific bound.

Soft-max Relaxation: Reviewer 3 asked for intuition on the $\tau$ parameter in the soft-max relaxation.Resolution: The authors demonstrated that as $\tau \to 0$, the method recovers the standard Condorcet winner (majority preference), while $\tau \to \infty$ approaches uniform proportional representation, effectively bridging RLHF and the proposed method.

**Reviewer Scores:**

Reviewer JTvB 4-> 6
Reviewer GtAR 2 -> 6

---

### Decision · Program_Chairs · 2026-01-26

Accept (Poster)